# Kaposi's sarcoma-associated herpesvirus induces specialised ribosomes to efficiently translate viral lytic mRNAs

James C. Murphy [1,2,5], Elena M. Harrington[1,2,5], Sophie Schumann[1,2,5], Elton J. R. Vasconcelos [3], Timothy J. Mottram [1,2], Katherine L. Harper[1,2], Julie L. Aspden [1,2,3] & Adrian Whitehouse [1,2,3,4] ✉

Historically, ribosomes were viewed as unchanged homogeneous macro-molecular machines with no regulatory capacity for mRNA translation. An emerging concept is that heterogeneity of ribosomal composition exists, exerting a regulatory function or specificity in translational control. This is supported by recent discoveries identifying compositionally distinct specialised ribosomes that actively regulate mRNA translation. Viruses lack their own translational machinery and impose high translational demands on the host during replication. We explore the possibility that KSHV manipulates ribosome biogenesis producing specialised ribosomes which preferentially translate viral transcripts. Quantitative proteomic analysis identified changes in the stoichiometry and composition of precursor ribosomal complexes during the switch from latent to lytic replication. We demonstrate the enhanced association of ribosomal biogenesis factors BUD23 and NOC4L, and the KSHV ORF11 protein, with small ribosomal subunit precursor complexes during lytic replication. BUD23 depletion resulted in significantly reduced viral gene expression, culminating in dramatic reduction of infectious virion production. Ribosome profiling demonstrated BUD23 is essential for reduced association of ribosomes with KSHV uORFs in late lytic genes, required for the efficient translation of the downstream coding sequence. Results provide mechanistic insights into KSHV-mediated manipulation of cellular ribosome composition inducing a population of specialised ribosomes facilitating efficient translation of viral mRNAs.

Regulation of gene expression is controlled through a vast number of mechanisms, from mRNA transcription through to protein turnover. At the level of mRNA translation, protein expression can be regulated through: the binding of proteins to untranslated regions (UTRs), upstream open reading frames (uORFs), structures within the mRNA itself, as well as lncRNA-mRNA base pairing[1–4]. Ribosome specialisation offers an additional route for the regulation of gene expression whereby distinct populations of ribosomes can preferentially translate specific mRNAs. Specialised ribosomes have been most extensively studied during development, however they have also been reported to

[1]School of Molecular and Cellular Biology, Faculty of Biological Sciences, University of Leeds, Leeds LS2 9JT, UK. [2]Astbury Centre for Structural Molecular Biology, University of Leeds, Leeds LS2 9JT, UK. [3]LeedsOmics, University of Leeds, Leeds LS2 9JT, UK. [4]Department of Biochemistry & Microbiology, Rhodes University, Grahamstown 6140, South Africa. [5]These authors contributed equally: James C. Murphy, Elena M. Harrington, Sophie Schumann. ✉ e-mail: a.whitehouse@leeds.ac.uk

have roles in cell type specificity, cell homoeostasis, and are implicated in various diseases[5–9]. Mechanisms leading to ribosome specialisation include: changes in ribosomal protein stoichiometry, paralogue switching, posttranslational modifications, addition of ribosome-associated proteins and changes in rRNA genetic variation and modifications[10].

Viruses must co-opt the host cell translation machinery to produce viral proteins. Viruses achieve this level of translational control in a variety of ways, for example many viruses utilise internal ribosome entry sites (IRES) to initiate translation in an m$^7$G cap independent manner[11,12]. Alternatively, viruses could manipulate the production of specialised ribosome populations, creating virus-specific specialised ribosomes to enhance the translation of their own viral mRNAs. This hypothesis is supported by a poxvirus kinase, which specifically phosphorylates serine/threonine residues in the human small ribosomal subunit protein, RACK1, dictating ribosome selectivity towards viral RNAs with 5′-UTR polyA-leaders[5,13].

Kaposi's sarcoma-associated herpesvirus (KSHV) is a large double stranded DNA virus associated with Kaposi's sarcoma and two lymphoproliferative disorders: primary effusion lymphoma and multicentric Castleman's disease[14]. Like all herpesviruses, KSHV has a biphasic life cycle comprising latent and lytic replication programmes. KSHV lytic replication and the efficient production of new virions is integral to disease progression and dissemination[15]. During lytic replication a temporal cascade of gene expression occurs requiring a great translational demand on the host cell compared to the relative dormancy of latency[16]. The lytic cascade therefore provides an opportunity for KSHV to impact the cellular ribosome population creating virus-specific specialised ribosomes that could efficiently translate viral mRNAs.

Ribosome biogenesis is a highly orchestrated process requiring the coordinated activity of all three RNA polymerases and over 400 transiently associated ribosome biogenesis factors (RBFs)[17]. Biogenesis begins in the nucleolus where precursor rRNA (pre-rRNA) is transcribed and assembled into a 90 S particle containing ribosomal proteins, RBFs and snoRNAs[18–22]. As ribosomal biogenesis progresses, endo- and exonucleolytic processing of the pre-rRNA separates the 90 S particle into a pre-40S and pre-60S particle, which are translocated from the nucleolus to the nucleoplasm[23]. Further maturation and checkpoint steps occur in the nucleoplasm before nuclear export and final cytoplasmic processing steps resulting in the removal of all RBFs yielding mature, translational competent 40 S and 60 S ribosomal subunits[24–26]. Throughout ribosome biogenesis the dynamic association of many RBFs can result in changes to the final mature ribosome, such as stoichiometric changes to rRNA modifications[10,27–29]. Importantly, mutations in these RBFs can lead to diseases such as specific cancers and tissue specific ribosomopathies[29–32].

Due to the relatively long half-life of ribosomes, changes to the cellular ribosome population during KSHV lytic replication were investigated by analysing differences in pre-40S ribosomal complexes undergoing biogenesis[24,33]. Herein we used quantitative proteomic affinity pulldowns to identify changes in the stoichiometry and composition of pre-40S ribosomal complexes between latent and lytic KSHV replication cycles. Biochemical analysis confirmed the increased association of the RBFs BUD23 and NOC4L with pre-40S ribosomal complexes during KSHV lytic replication and their essential role for lytic replication. We further identified that the previously uncharacterised KSHV ORF11 protein associates with pre-40S ribosomal subunits and is essential for the production of new infectious virions. Using ribosome profiling we precisely identify changes in the translation of the KSHV transcriptome during lytic replication that are dependent on BUD23. Notably, we show that BUD23 is essential for the reduced translation of KSHV mRNA uORFs, by virus-specific specialised ribosomes, from genes encoding viral proteins involved in the late lytic gene cascade. Together these results describe a novel

mechanism by which KSHV engineers the host translational machinery to produce specialised ribosomes that efficiently translate late lytic KSHV mRNAs.

## Results

### Changes to pre-40S ribosomal complexes during KSHV lytic replication determined by quantitative mass spectrometry

To characterise changes to pre-40S ribosomal complexes during KSHV infection, we first established stable cell lines of TREx BCBL1-Rta cells expressing FLAG-2xStrep-tagged ribosome biogenesis proteins. Tagged bait proteins DIMT1, PNO1, TSR1 and LTV1, were selected due to their temporal and spatial association to the pre-40S ribosomal complex covering the maturation of the 18 S rRNA (Fig. 1a). Affinity pulldowns with bait proteins were performed in latent KSHV-infected cells and compared with cells after 24 h lytic reactivation, then analysed by quantitative mass spectrometry (Fig. 1b–f; Supplementary Fig.1–4, and Supplementary Data 1). Changes in bait protein-associated pre-40S complexes were identified by comparing total net abundance of proteins between KSHV latent and lytic samples (Fig. 1c–f) and by comparing fold change between the two conditions (Fig. 1g).

Together these results show dynamic changes in the stoichiometry of associated ribosomal biogenesis factors between latent and lytic replication cycles, especially at later stages of biogenesis. To identify the most confident hits, we investigated proteins with the largest changes in association between latent and lytic replication cycles, combined with changes that were observed in pulldowns of multiple bait proteins. Using these criteria, we identified two clusters of associated RBFs that function cooperatively: BUD23 and TRMT112, as well as NOC4L, NOP14 and EMG1. BUD23 is a methyltransferase, which mediates the N$^7$-methylation of G1639 in the 18 S rRNA, and TRMT112 acts as a co-factor for BUD23 and other RBFs[34]. NOC4L, NOP14, EMG1 and UTP14A form a complex which associates with pre-40S ribosome complexes in the nucleolus[35]. Their function during ribosome biogenesis is not fully understood, however EMG1 also functions as a methyltransferase, which catalyses the N$^1$-methylation of U1240 in the 18 S rRNA[36]. Notably, the previously uncharacterised KSHV protein, ORF11, was also shown to associate with pre-40S ribosomal complexes during lytic replication (Fig. 1g).

### BUD23 has increased association with pre-40S ribosomal complexes and is required for efficient KSHV lytic replication

To confirm the quantitative proteomic results, we repeated the pulldown experiments using FLAG-2xStrep-PNO1 as the bait protein in TREx BCBL1-Rta cells during latency or lytic replication, as this was the complex with the largest enrichment for BUD23 and NOC4L. Pulldowns were analysed by western blotting, probing with antibodies against BUD23, NOC4L, as well as eS19, a protein for which no changes were detected by mass spectrometry (Fig. 1). These data confirmed an increased association of BUD23 (4.3-fold) and NOC4L (3.1-fold) to pre-40S complexes during KSHV lytic replication, in contrast no change was observed for the core ribosomal protein eS19 (Fig. 2a, b). Interestingly, total cellular levels of BUD23 are not altered and NOC4L decreases during KSHV lytic replication, suggesting that the increased association of both BUD23 and NOC4L is directly driven by the virus (Supplementary Fig. 5). To validate the integrity of isolated PNO1 pre-40S complexes, we examined their rRNA content through isolation of total RNA and analysis by agarose gel electrophoresis and qPCR. Both show the isolated PNO1 pre-40S complexes contain highly pure 18 S rRNA (Supplementary Fig. 6).

To investigate the role of BUD23 during KSHV lytic replication, BUD23 stable knockdown cell lines were produced in TREx BCBL1-Rta cells transduced with a variety of shRNAs. Effective BUD23 knockdown was achieved using two different shRNAs, reducing BUD23 mRNA levels by 83% and 84%, and protein levels by 83% and 73%, respectively (Fig. 2c, d). Depletion of BUD23 had no effect on cell viability, as

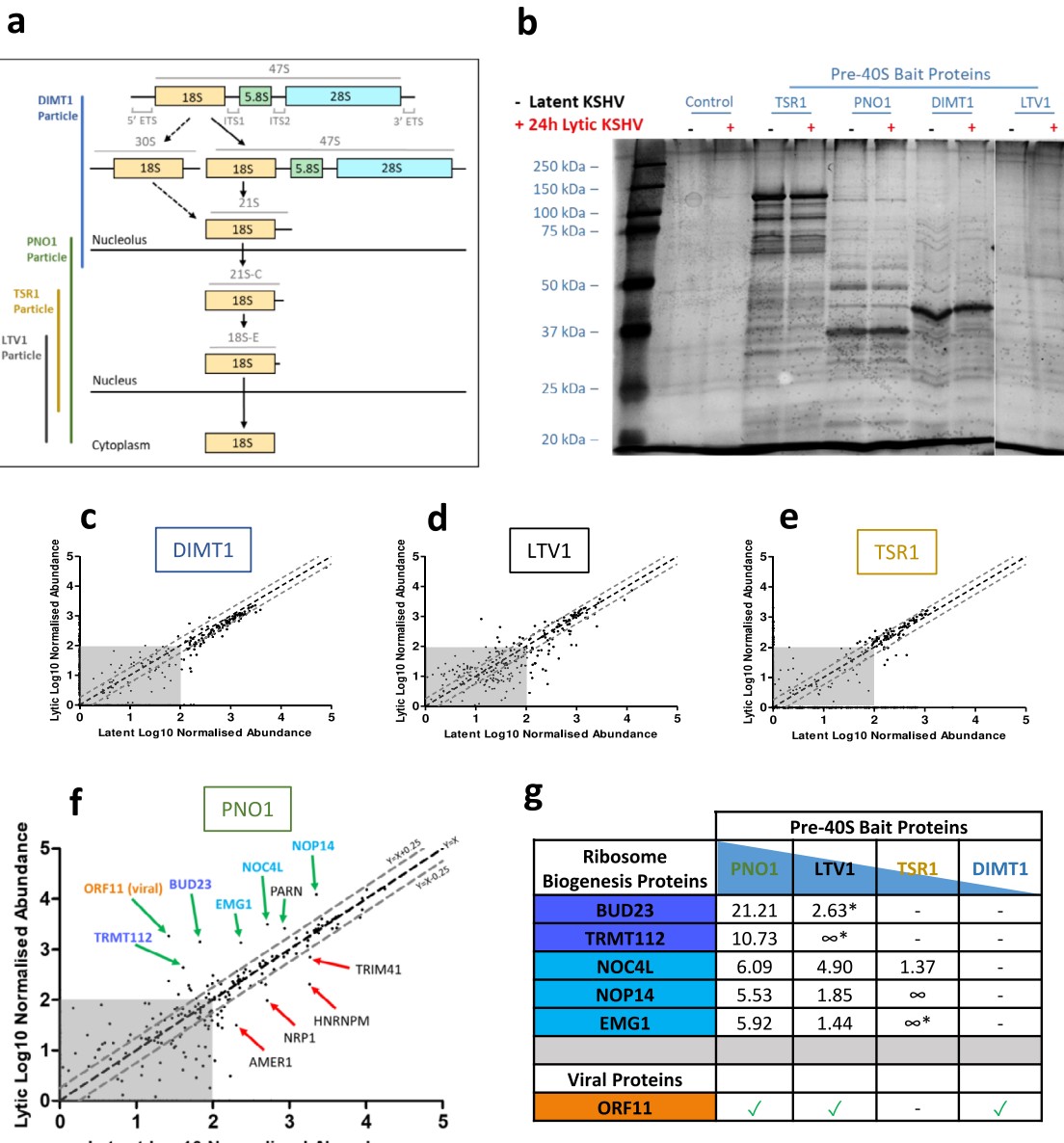

**Fig. 1 | Changes to pre-40S ribosomal complexes during lytic KSHV infection determined by quantitative mass spectrometry. a** Temporal Association of the 40 S ribosome biogenesis factor bait proteins during the maturation of the 18 S rRNA. **b** Whole cell lysates from latent and 24 h post lytic reactivation control TREx BCBL1-Rta cell line or TREx BCBL1-Rta cell lines stably expressing pre-ribosome biogenesis bait proteins were subject to Twin-Strep-tag® pulldowns. Silver stain gels of isolated pre-40S ribosome complexes. Representative stained gel is shown (*n* = 2 biologically independent samples). **c–f** TMT LC-MS/MS was used to identify the association and stoichiometry of all proteins present in the isolated complexes.

Incorporation of each protein into the pre-40S ribosomal complexes isolated from cells infected with latent KSHV (*x*-axis) compared to lytic KSHV (*y*-axis) A minimum cut-off for incorporation was set to 100 abundance indicated by the grey boxes. A minimum threshold for increase and decrease incorporation of proteins into complexes isolated from lytic KSHV cells was set using the equations $Y = X + 0.25$ and $Y = X - 0.25$ respectively. **g** Ratio increases in incorporation of two select groups of ribosome biogenesis in pre-40S ribosomal complexes isolated from cells infected with lytic KSHV compared to latent KSHV ∞ = only detected in lytic pre-40S complexes. * = below 100 abundance.

measured by global translation, proliferation rate and metabolic activity (Fig. 2e, f and Supplementary Fig. 7). Furthermore, BUD23-depleted cells produced equal levels of 80 S ribosomes and polysomes compared to scrambled controls, indicating that loss of BUD23 does not negatively affect ribosome biogenesis or global translation in KSHV-latently infected cells (Fig. 2g). However, the catalytic function of BUD23 to mediate the N[7]-methylation of 18 S rRNA G1639 was significantly reduced, by up to 37%, in BUD23-depleted cells compared to basal levels in scrambled control cells (Fig. 2h).

To allow sufficient time for KSHV to impact the host cell ribosome population during lytic replication it is likely that any changes introduced to ribosomes upon reactivation would be caused by early or

delayed early viral proteins and impact proteins expressed later during the lytic expression cascade (intermediate or late genes). As such, we quantified protein expression of early and delayed early viral proteins, ORF57 and ORF59 respectively, in the BUD23-depleted TREx BCBL1-Rta cells versus scrambled control cells (Fig. 3a–c and Supplementary Fig. 8a, b). As expected, similar levels of both ORF57 and ORF59 mRNA and protein were observed in scrambled control and BUD23-depleted cells throughout a 48 h time course of lytic reactivation. These data were further exemplified by qPCR analysis of highly active translating ORF57 and ORF59 mRNAs, which were isolated from polysome profiles, showing no change between BUD23 depletion and scrambled control cells (Supplementary Fig. 9a, b). In contrast, quantification of

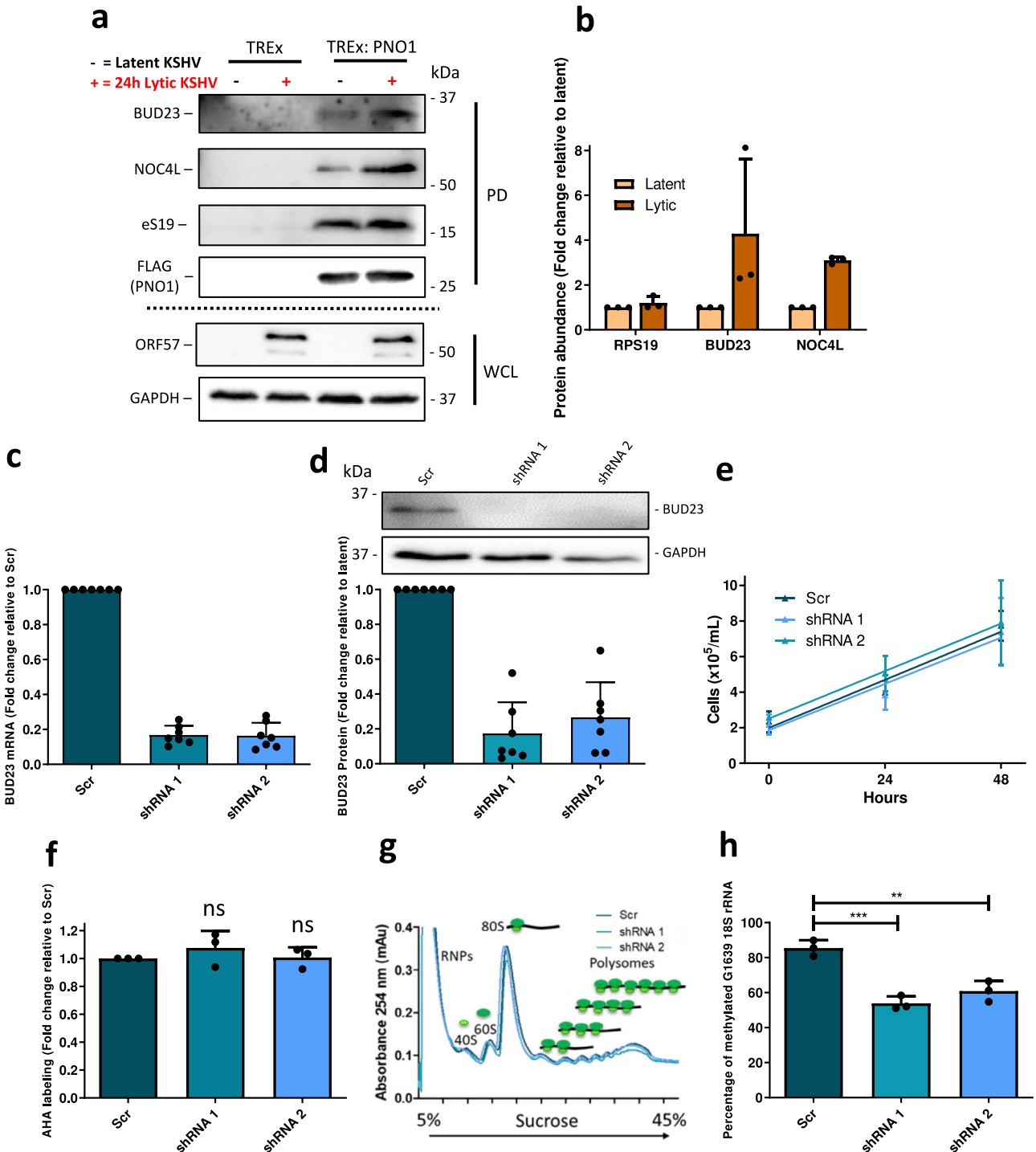

late structural viral proteins, K8.1 and ORF65, showed a dramatic reduction in proteins levels of up to 67% and 73% respectively, in BUD23-depleted cells compared to scrambled control cells at 48 h post lytic reactivation (Fig. 3a, d, e). In contrast, we only observed a slight reduction in mRNA levels of 32% and 25% for K8.1 and ORF65 respectively in BUD23-depleted cells compared to scrambled control cells at 48 h post-reactivation (Supplementary Fig. 8c, d). Furthermore, depletion of BUD23 results in significant reduction of highly active translating K8.1 and ORF65 mRNAs, by up to 69% and 66% respectively, isolated from polysome profiles compared to scrambled control cells (Supplementary Fig. 9c, d). These data demonstrate that depletion of BUD23 greatly reduces the ability of KSHV to progress though the late lytic gene cascade, which ultimately results in a dramatic reduction in

infectious virion production of up to 91% (Fig. 3f). This was further demonstrated by the almost total loss of staining for the viral marker LANA in naive HEK-293T cells which were re-infected with virus produced from cells depleted of BUD23 (Supplementary Fig. 10). These results demonstrate that BUD23 depletion leads to the breakdown of the KSHV late lytic gene cascade. This breakdown causes a significant reduction in the expression of late lytic genes, with a limited extent on transcription compared to a dramatic effect at the translational level. Overall, the depletion of BUD23 leads to the decrease in expression of late structural KSHV proteins which prevents the formation of new infectious virions.

Finally, to examine the role of BUD23-mediated G1639 methylation during KSHV lytic replication we analysed the $N^7$-methylation of

**Fig. 2 | BUD23 has increased association with pre-40S ribosomal complexes during lytic replication but stable knockdown is dispensable for normal cell functions. a** Latent and 24 h post reactivation whole cell lysates from a control TREx BCBL1-Rta cell line or a TREx BCBL1-Rta cell line stably expressing a PNO1 bait protein were subject to Twin-Strep-tag® pulldowns. Representative western blots from pulldowns (PD) and whole cell lysate (WCL) inputs (*n* = 3 biologically independent samples). **b** Densitometric analysis of pulldown western blots normalised to bait protein PNO1 (*n* = 3 biologically independent samples). **c** A Lentivirus expression system was used to stably transduce TREx BCBL1-Rta cells with a non-targeting scrambled shRNA (Scr) or two different shRNAs targeting BUD23 (shRNA 1 and 2). BUD23 mRNA production was assayed by two step RT-qPCR and analysed by comparison to the Scr control using a ΔΔCt method (*n* = 7 biologically independent samples). **d** Whole cell lysates were collected and analysed by western blot probing for BUD23, GAPDH was included as a reference gene, representative western blots and densitometric analysis relative to the Scr control (*n* = 7 biologically independent samples). **e** BUD23 cell lines were counted over 48 h to measure cell proliferation (*n* = 4 biologically independent samples). **f** Global translation was determined by depleting cells of methionine for 60 min and then nascent peptides were labelled with click-iT® L-Azidohomoalanine (AHA). Finally, cells were fixed, permeabilised and stained with a Click-iT® reaction using an alkyne Alexa Fluor™ 488 and quantified using flow cytometry (*n* = 3 biologically independent samples). **g** The gross ribosome population in the cells was determined by polysome profiling (RNPs = ribonuclear proteins). **h** Total RNA was isolated from BUD23 knockdown and scrambled cell lines, and chemically cleaved at $m^7G$ sites. The proportion of cleaved to uncleaved rRNA at 18 S G1639 was determined by qPCR with primers flanking the cleave site and analysed using a ΔΔCt method *p* = 0.0004 and 0.0014 (*n* = 3 biologically independent samples). Data are presented as mean ± SD. Significance was calculated by one-way analysis of variance (ANOVA) with a Newman–Keuls multiple comparison post-test. Asterisks denote a significant difference between the specified groups (**p* ≤ 0.05, ***p* < 0.01 and ****p* < 0.001).

the 18 S rRNA base G1639 over a 48 h reactivation time course. Notably, 18 S rRNA G1639 methylation increases by almost 10% during KSHV lytic replication and remains significantly reduced when BUD23 is depleted (Fig. 3g and Supplementary Fig. 11). This, taken together with the dramatic effect of BUD23 knockdown and associated reduction of G1639 methylation on KSHV lytic replication, suggests the $N^7$-methylation of G1639 may be required for the effective translation of late KSHV genes or genes involved in late gene expression.

### The KSHV ORF11 protein interacts with pre-ribosome complexes during lytic replication

Our quantitative proteomic analysis of isolated pre-40S ribosome complexes using PNO1, LTV1 and DIMT1 bait proteins identified one specific KSHV lytic protein, ORF11, which co-precipitated with high abundance (Fig. 1). We hypothesised that ORF11 might form part of the mechanism by which KSHV modulates ribosome biogenesis and regulates changes in association of cellular factors. Therefore, to characterise the role of ORF11, we first confirmed its interaction with pre-40S ribosome complexes by reverse pulldown analysis using GFP-tagged ORF11 stably expressed in TREx BCBL1-Rta cells. Western blot analysis of the GFP-trap bead precipitates confirmed ORF11 interacts with pre-ribosomal and ribosomal proteins of the 40 S subunit (Fig. 4a). These interactions were not affected by induction of KSHV lytic replication in cells overexpressing GFP-ORF11 or GFP alone. Moreover, the interaction between ORF11 and pre-40S ribosome complexes is dependent on the presence of rRNA (Supplementary Fig. 12). However, which pre-ribosomal proteins ORF11 interacts with directly is unknown at present. Finally, we also observed ORF11 interacting with a large ribosomal subunit protein uL23 which might be due to the closely linked final cytoplasmic maturation steps of the pre-40S and 60 S subunits.

To confirm whether ORF11 localises to sites of ribosome biogenesis, we generated a TREx BCBL1-Rta cell line which stably expressed ORF11-FLAG to assess its localisation by confocal microscopy. ORF11-FLAG was found to mainly reside at known sites of ribosome biogenesis, such as the nucleus and nucleolus where it co-localises with the RBF, BUD23 (Supplementary Fig. 13a, b). These data further point to an intimate role of ORF11 with ribosome biogenesis.

To conclusively determine if ORF11 associates with both pre-40S and mature 40 S ribosome subunits we performed polysome profiling on TREx BCBL1-Rta cells stably expressing GFP-ORF11. Western blot analysis of polysome profile fractions demonstrated that GFP-ORF11 associated with these ribosomal complexes (Fig. 4b). To further understand the interactions of ORF11, GFP affinity pulldowns of GFP-ORF11 compared to a control GFP pulldown were analysed by quantitative proteomics. Of the 129 proteins identified to associate with GFP-ORF11, 83 were ribosomal proteins or ribosome biogenesis factors, again demonstrating the intimate relationship of ORF11 with ribosomes and ribosome biogenesis (Fig. 4c and Supplementary Data 2). Two of

the non-ribosome associated proteins, which interact with GFP-ORF11 were from the importin α family, suggesting a potential mechanism for ORF11 nuclear import, allowing its interaction with pre-40S ribosome subunits during biogenesis (Supplementary Data 2). Importin α interactions were further validated by immunoprecipitations, with western blot analysis of the precipitates demonstrating the interaction of ORF11 with importin α1 but not α5 (Supplementary Fig. 14).

To gain a greater understanding of how ORF11 interacts with pre-40S ribosomal subunits during biogenesis, we mapped the abundance of the small subunit ribosomal proteins identified from the quantitative proteomic affinity pulldowns to the structure of the pre-40S ribosomal subunit (Plassart et al. 2021, PDB: 6ZUO - https://www.rcsb.org/structure/6ZUO)[37]. The most highly abundant 40 S ribosomal proteins interacting with GFP-ORF11 localise around the rRNA on the subunit interface side of the pre-40S ribosomal subunit, which include eS30, uS19, uS13, eS19, eS8, and uS15 (Fig. 4d). Interestingly, this is also the location where BUD23 interacts with the rRNA and where the 18 S rRNA base G1639 is located. We therefore speculate that the rRNA at the subunit interface is potentially the region ORF11 interacts with the pre-40S ribosome.

Finally, to confirm the essential role of ORF11 in the KSHV lytic replication cycle and also to determine whether depletion of ORF11 resulted in a similar phenotype observed with depletion of the cellular ribosome biogenesis factor BUD23, we generated CRISPR/Cas9 ORF11 knockout cell lines. Specifically, mutations were generated in the KSHV genome creating premature STOP codons in the ORF11 gene using two different gRNAs, resulting in either 24 amino acid (gRNA 1) or 57 amino acid (gRNA 2) truncated versions of ORF11, which normally consists of 407 amino acids. Protein expression of the early ORF57 and late ORF65 viral proteins were then quantified after 24 and 48 h of lytic infection, respectively (Fig. 4f). As expected, there was a dramatic reduction in ORF65 protein expression compared to the scr control, whereas early ORF57 protein expression was still evident, with only a small reduction in ORF57 levels observed. To confirm the effect was due to ORF11 knockdown, an ORF11 overexpression construct was stably transduced into the gRNA 1 cell line. Figure 4F shows that recovering ORF11 expression allowed for recovery of both early and late protein levels, with ORF57 returning to 89% expression, and ORF65 to 86% expression compared to the scr control. Virus reinfection assays were then used to assess the ability of ORF11 mutated viruses to produce new infectious virions (Fig. 4e and Supplementary Fig. 15). Results indicate a significant reduction in infectious virion production after ORF11 knockout, similar to that of BUD23. This confirms that ORF11 is essential for effective KSHV lytic replication.

### KSHV specialised ribosomes promote the efficient translation of KSHV late lytic genes which contain uORFs

To dissect the impact of BUD23 depletion on the translation of KSHV mRNAs we used ribosome profiling to screen global translation during

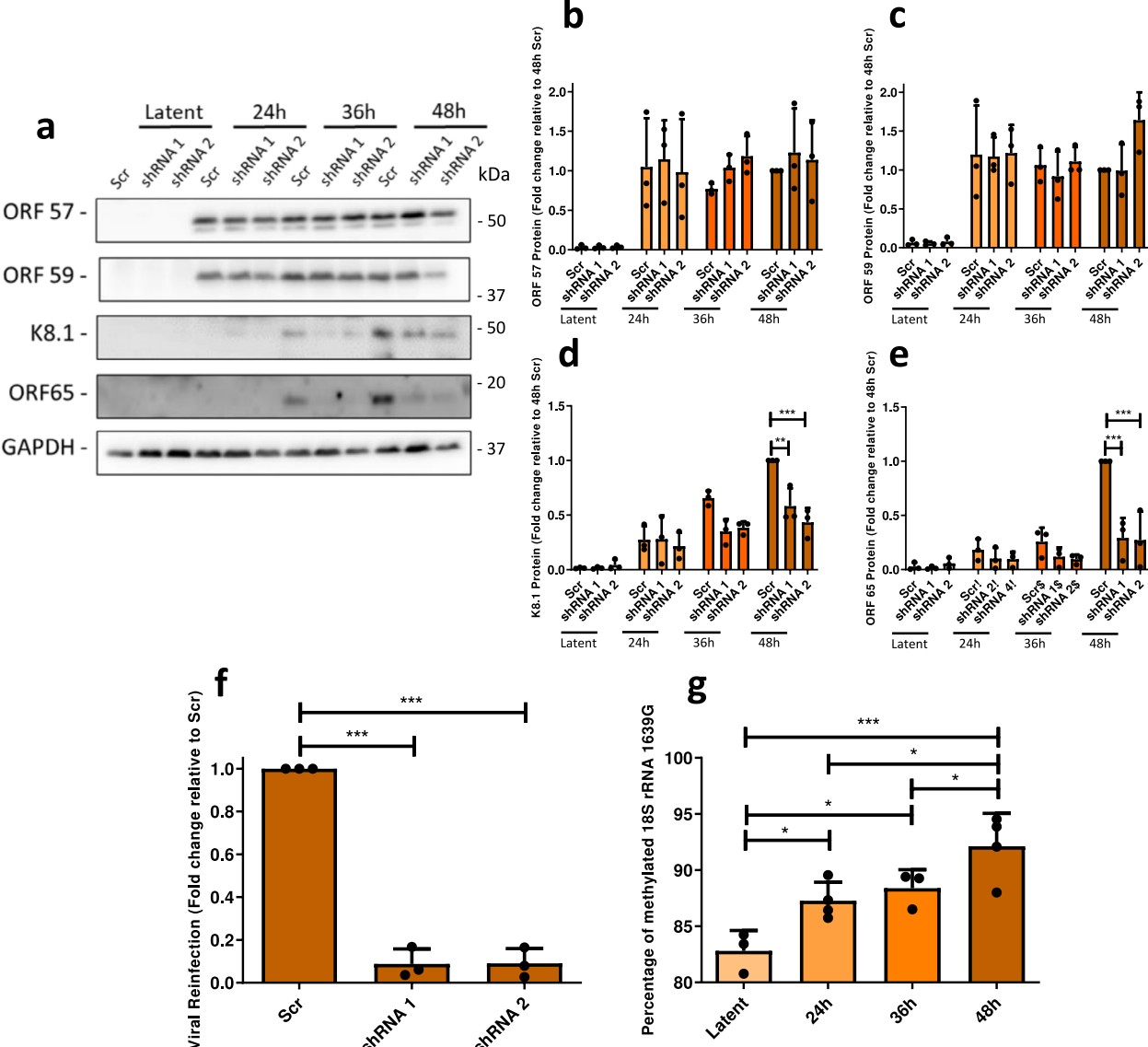

**Fig. 3 | Depletion of BUD23 does not affect early KSHV lytic replication but significantly impacts the late stages.** TREx BCBL1-Rta cells expressing a Scr shRNA or two different shRNAs targeting BUD23 were used for a time course of KSHV lytic reactivation over 48 h, with samples collected at 0 (latent), 24, 36 and 48 h (*n* = 3 biologically independent samples). **a** Whole cell lysates were collected and analysed by western blot probing early KSHV lytic proteins, ORF57 and ORF59, late KSHV lytic proteins, K8.1 and ORF65, and GAPDH was included as a loading control. **b**–**e** Densitometric analysis of ORF57, ORF59, K8.1 (*p* = 0.0045 and <0.0001) and ORF65 (*p* ≤ 0.0001 for both) relative to GAPDH are shown. **f** Lytic reactivation of KSHV was induced in TREx BCBL1-Rta cells expressing a Scr shRNA or two different shRNAs targeting BUD23 for 72 h. Virus released from TREx BCBL1-Rta cell lines was collected and HEK 293T cells re-infected with the virus for 48 h.

Total RNA was isolated from cells and quantified by two-step RT-qPCR, with primers specific for the early lytic viral gene ORF57 (*p* ≤ 0.0001 for both) (*n* = 3 biologically independent samples). Data was analysed by comparison to GAPDH and the Scr control using a ΔΔCt method. **g** Total RNA was isolated from TREx BCBL1-Rta cells at 0 (latent), 24, 36 and 48 h post reactivation and chemically cleaved at m7G sites. The proportion of cleaved to uncleaved rRNA at 18 S G1639 was determined by qPCR with primers flanking the cleave site and analysed using a ΔΔCt method (*n* = 3 biologically independent samples). Data are presented as mean ± SD. Significance was calculated by one-way ANOVA with a Newman–Keuls multiple comparison post-test. Asterisks denote a significant difference between the specified groups (**p* ≤ 0.05, ***p* < 0.01 and ****p* < 0.001).

infection. Scrambled control TREx BCBL1-Rta cells were compared to BUD23 depleted cells, at 36 h post lytic KSHV reactivation (Fig. 2). Ribosome profiling identified the increased translation of numerous uORFs within KSHV late lytic genes, upon BUD23 depletion, including uORFs within the 5'-UTRs of ORF69, 75 and 61 (Fig. 5a, b). An increase in the association of ribosomes with a uORF typically leads to a reduction in translation of the downstream main coding sequence (CDS)[1]. We detected such a reduction in translation efficiency and ribosome occupancy of the CDS for the late lytic KSHV gene ORF28, which also contains a uORF in its 5'-UTR (Fig. 5a, b). Translation of KSHV uORFs with the greatest increase in translational efficiency, during BUD23

depletion, are significantly increased compared to the translation of their corresponding CDS (Supplementary Fig. 16). Interestingly, depletion of BUD23 did not significantly affect the translation efficiency of any human genes (Supplementary Fig. 17). These data suggest that BUD23 is required for the correct regulation of late lytic KSHV uORFs and therefore the efficient translation of downstream CDSs.

To test this hypothesis, we developed a luciferase reporter assay where KSHV 5'-UTRs containing uORFs were cloned upstream of a luciferase reporter and as a control the start codon of each uORF was mutated by site directed mutagenesis (Fig. 5c). Mutation of the uORF start codon in KSHV 5'-UTRs from genes ORF69, 28, and

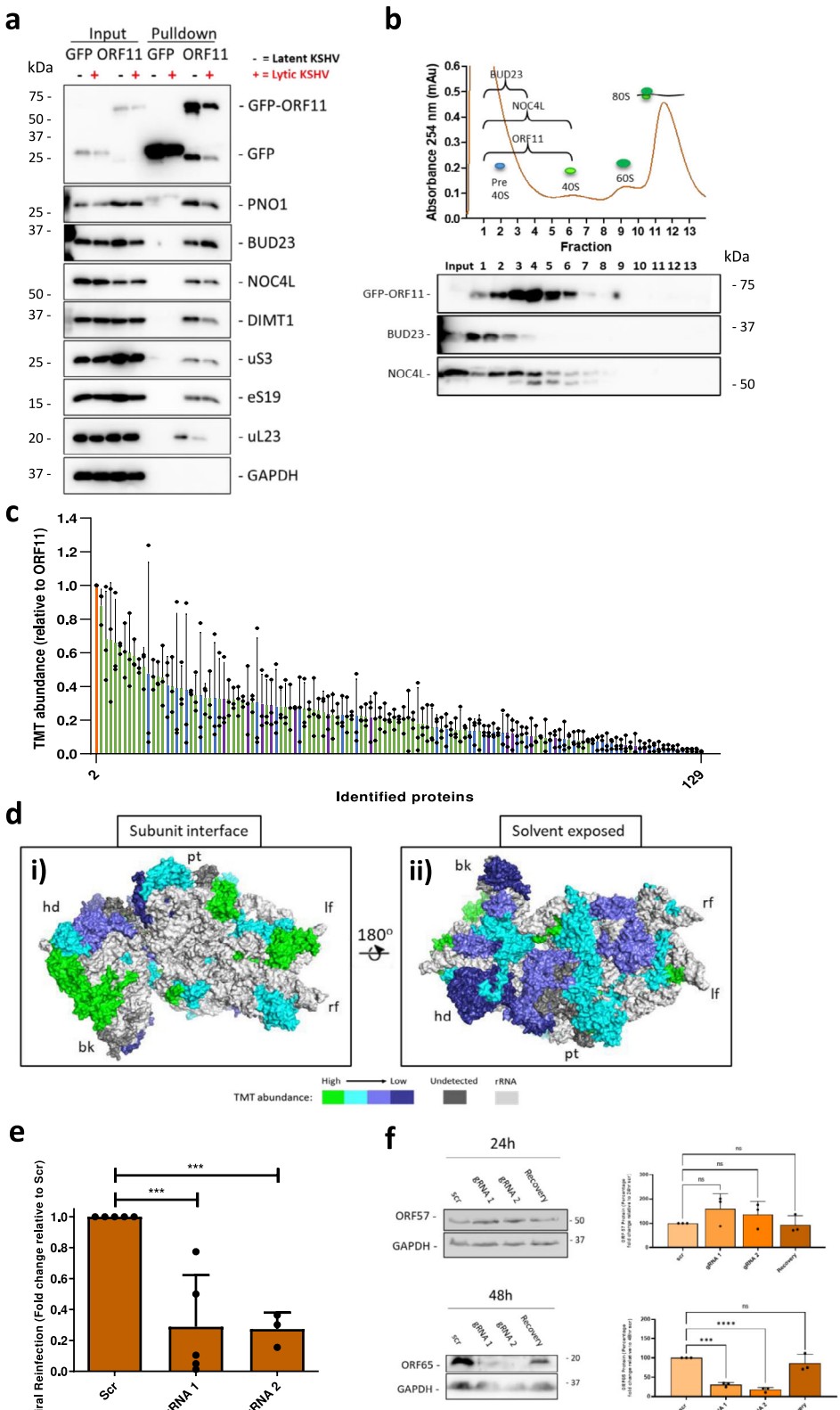

34 all significantly increased the expression of a downstream luciferase reporter (Fig. 5d), demonstrating that these uORFs negatively regulate the translation of a downstream CDS. However, no significant change in luciferase expression was observed for the KSHV 5'-UTRs containing uORFs of genes ORF65 and 30 (Supplementary Fig. 18). The mechanism and/or function of different KSHV uORF groups may therefore vary as the presence of ORF65 and ORF30

uORFs in the 5'-UTR do not directly impact the downstream expression of a CDS.

To determine the role BUD23 and ORF11 play in regulating the translation of these uORF-reporters, we created HEK-293T cells which stably expressed either a scrambled shRNA control, shRNAs targeting BUD23 or an overexpression construct for ORF11-FLAG (Supplementary Figs. 19, 20). We confirmed that the proliferation of each cell line

**Fig. 4 | KSHV ORF11 associates with pre-40S ribosomal complexes and is essential for lytic replication. a** Latent and 24 h reactivated whole cell lysates from TREx BCBL1-Rta GFP control or TREx BCBL1-Rta GFP-ORF11 cells were subject to GFP-Trap® pulldowns. Representative western blots from pulldowns and inputs (*n* = 3 biologically independent samples). **b** Polysome profile of TREx BCBL1-Rta GFP-ORF11 cells (bi). Representative western blots of polysome profile sucrose fractions probing for GFP, BUD23 and NOC4L (*n* = 3 biologically independent samples). (bii). **c** TMT LC-MS/MS identified the association and stoichiometry of proteins present in GFP-Trap® pulldowns from latent cells (*n* = 3 biologically independent samples). Minimum cut-off for incorporation was set to 1% abundance of GFP-ORF11 bait and a > 1.5 fold increase over control GFP pulldown background. Average TMT abundance values relative to GFP-ORF11 bait for proteins above the minimum cut-off values. **d** Abundance of small subunit ribosomal proteins in GFP-ORF11 pulldowns ranked as fold change over GFP background, ≥7 (green), 6–7 (cyan), 5–6 (light blue), 2–5 (dark blue), and not detected (dark grey). Pre-40S

ribosomal structure (PDB: 6ZUO). bk, beak; hd, head; lf, left foot; pt, platform; rf, right foot. **e** TREx BCBL1-Rta cells expressing a Scr gRNA or two ORF11-targeting gRNAs were reactivated for 72 h. Virus released was used to reinfect 293T cells for 48 h. Total RNA was isolated and quantified by two-step RT-qPCR, with ORF57-specific primers (*n* = 3 biologically independent samples) (*p* = 0.0008 and 0.0010). **f** TREx BCBL1-Rta cells expressing a Scr gRNA, two ORF11-targeting gRNAs and ORF11 overexpression plasmid (recovery) were used to assess KSHV lytic reactivation over 24 and 48 h (*n* = 3 biologically independent samples) (*p* = 0.0003 and <0.0001). Whole cell lysates were analysed by western blot probing KSHV early lytic protein ORF57, late lytic protein ORF65, and GAPDH as a loading control. Representative western blots and densitometric analysis of ORF57 and ORF65 relative to GAPDH (*n* = 3 biologically independent samples). **f** Data are presented as mean ± SD. Significance was calculated by one-way ANOVA with a Newman–Keuls multiple comparison post-test. Asterisks denote a significant difference between the specified groups (*\*p* ≤ 0.05, *\*\*p* < 0.01 and *\*\*\*p* < 0.001).

was not affected by the expression of either BUD23 targeting shRNAs or the over-expression of ORF11-FLAG (Supplementary Fig. 21). Notably, BUD23 depletion significantly reduced the expression of a downstream luciferase reporter in the presence of KSHV 5'-UTRs containing uORFs of genes ORF69, 28 and 34 compared to a scrambled control cell line (Fig. 5e–g). Conversely, ORF11 overexpression resulted in an increase in downstream luciferase reporter expression (Fig. 5e–g). Importantly, the 5'UTR of ORF69, 34 and 28 do not affect the transcription of the luciferase reporter gene during BUD23 depletion or with expression of ORF11 (Supplementary Fig. 22). These data corroborate the hypothesis that the function of BUD23 is required for the efficient translation of various KSHV late genes, which contain uORFs. Furthermore, the KSHV ORF11 protein enhances the translation of these CDSs most likely though facilitating the increased association of BUD23 with pre-40S ribosomal subunits during biogenesis. Together these results suggest 40 S ribosomal subunits containing the m⁷G1639 modification reduce the translation of specific uORFs present in KSHV late lytic transcripts, which in turn enhances the translation of the downstream CDSs.

## Discussion

Viruses lack their own translational machinery and must manipulate the host cell environment to optimise production of new infectious virions. It would therefore seem highly likely that some viruses manipulate the production of specialised ribosomes to efficiently translate specific viral transcripts. In this study we highlight a novel mechanism utilised by KSHV to induce changes during ribosome biogenesis which facilitate formation of virus-specific specialised ribosomes to enhance viral protein translation during KSHV lytic replication.

Using tagged ribosome biogenesis proteins as bait, we discovered distinct alterations to the pre-40S ribosomal subunit during the biogenesis process. We observe an increased association of the methyltransferase BUD23 and its co-factor TRMT112, and the NOC4L-NOP14-EMG1 complex during KSHV lytic replication. Further investigation of the role of BUD23, using depletion studies, showed that it is dispensable for host cell proliferation, metabolic activity and the overall ribosome population in KSHV-latently infected cells, which correlates to previous observations in other cell lines[38]. Notably, BUD23 depletion does not affect the translation of KSHV early lytic transcripts, such as ORF57 and ORF59. However, BUD23 was found to be necessary for the expression of late viral transcripts and essential for the effective production of new infectious virions. Overall, depletion of BUD23 has a dramatic impact on the later stages of the KSHV lytic gene cascade, not surprisingly the point during lytic replication with the highest translational demand on the cell, in order to produce the vast quantity of structural proteins required for the formation of new virions.

Furthermore, we identified that the previously uncharacterised KSHV ORF11 protein associates with pre-ribosome complexes during

lytic replication. Interestingly, ORF11 was not found in TSR1 pulldowns, even though temporally, the interaction of TSR1 with the pre-40S complex is overlapping with both PNO1 and LTV1. TSR1 interacts with the grove in between the beak and the right foot of pre-40S subunits and as such may indicate that ORF11 binding is located further towards the head and beak of the subunit interface, supported by a high association of ribosomal proteins with ORF11 at this site[24]. We also observed an interaction of ORF11 with a large ribosomal subunit protein, uL23. While our data from western blotting of polysome fractions shows that the major interaction for ORF11 is with the pre-40S subunit, the final cytoplasmic maturation steps of the pre-40S and 60 S subunits are intimately intertwined, including quality control docking of mature 60S subunits with pre-40S subunits[25,39,40]. As ORF11 appears to interact with pre-40S subunits potentially up until maturity this could explain why some interaction with the 60S subunit is also observed. The transient nature of ORF11 association is further supported by immunofluorescence studies which highlight ORF11 nucleolar and nucleoplasm localisation, again emphasizing the role of ORF11 in ribosome biogenesis. We did not observe ORF11 localising with actively translating ribosomes in the cytoplasm suggesting the primary role of the ORF11 interaction is to reprogramme ribosomes in the nucleolus. Overall, our data suggests that during lytic replication ORF11 associates with pre-40S complexes to enhance the association of BUD23, and potentially other RBFs, leading to an increased population of virus-specific specialised ribosomes containing the 18 S rRNA m⁷G1639 modification (Fig. 6).

Using ribosome profiling and luciferase reporter assays we identified a novel mechanism through which the function of BUD23 and ORF11 impact the translation of late lytic KSHV genes. Our results suggest that BUD23 is required for reduced translation of uORFs in late lytic KSHV mRNAs and consequently the efficient translation of the downstream CDS (Fig. 6). Furthermore, ORF11, most likely by facilitating the increased association of BUD23 with pre-40S ribosomes subunits, has the same effect on translation of uORF containing late lytic KSHV mRNAs (Fig. 6). Interestingly, uORFs from the late lytic KSHV genes ORF65 and 30 had no impact on the translation of a downstream CDS. These uORFs have been identified in our ribosome profiling data and by Arias et al. but suggest their biological function differs to those from ORF69, 28 and 34 and do not act to repress translation[16]. The KSHV late lytic genes we identified that are impacted by BUD23 and ORF11 all have integral functions during KSHV late lytic replication; ORF69, nuclear egress; ORF28, immune evasion; and ORF34, a key component of the late gene transcription preinitiation complex[41–43]. Therefore, the reduction in translation of these and other genes would be highly detrimental to the KSHV late lytic gene cascade and production of new infectious virions.

The increased methylation of the 18 S rRNA base G1639 by BUD23 during KSHV lytic replication, mostly likely driven by ORF11-mediated increased association, could produce a population of ribosomes that

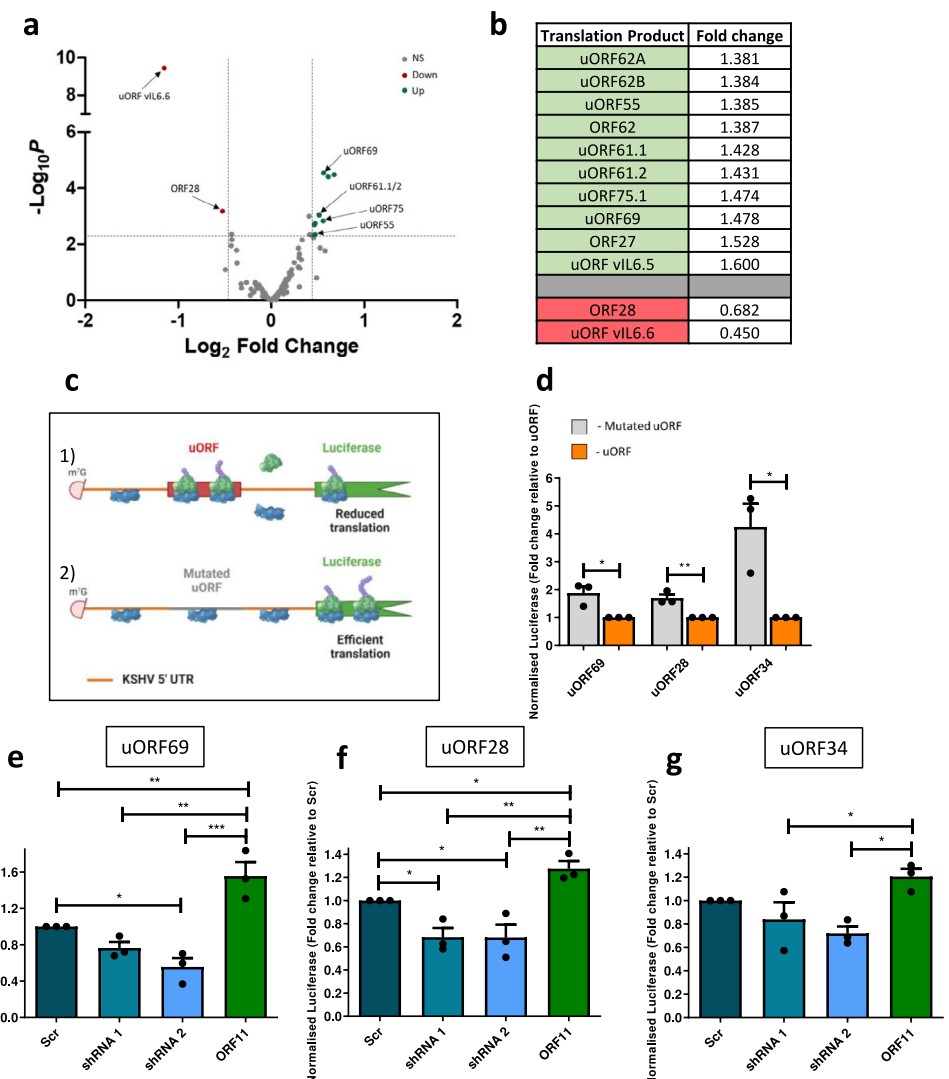

**Fig. 5 | ORF11 and BUD23 are required for the efficient translation of key late lytic KSHV mRNAs which contain uORFs. a** Changes in translational efficiency of viral genes calculated for libraries originating from TREx BCBL1-Rta cells expressing a Scr shRNA compared to shRNA 1 targeting BUD23 at 36 h post KSHV lytic reactivation. A false discovery rate was set at <0.005 and fold change threshold set at ±0.45 (log2). Volcano plot representation of all KSHV gene translational efficiency changes ($n = 2$ biologically independent samples). **b** Table of all KSHV genes with significantly changed translational efficiencies >± 0.45 (log$_2$). **c** Schematic of luciferase reporter assays with upstream KSHV mRNA 5' UTRs containing (1) uORFs or (2) with the uORF start codon mutated in HEK 293T cells. **d** Normalised luciferase intensity was quantified from luciferase reported plasmids containing uORFs or with the start codon of the uORF mutated ($n = 3$ biologically independent samples). Significance was calculated by using two-tailed unpaired $t$-tests ($p = 0.0208$, 0.0060 and 0.0177). **e–g** Normalised luciferase intensity of KSHV uORF containing reporter plasmids was quantified in HEK 293T cells stably expressing a non-targeting scrambled (Scr) shRNA or shRNAs 1 or 2 targeting BUD23 or ORF11-FLAG exogenous expression ($n = 3$ biologically independent samples). Data are presented as mean ± SD. Significance was calculated by one-way ANOVA with a Newman–Keuls multiple comparison post-test. Asterisks denote a significant difference between the specified groups (*$p ≤ 0.05$, **$p < 0.01$ and ***$p < 0.001$). NS Not significant.

are specialised for the efficient translation of key late lytic genes which contain uORFs (Fig. 6). This could provide an additional regulatory mechanism of late lytic gene expression, preventing expression until the desired time at later time points during lytic replication. As such, BUD23 and ORF11 could act as regulators of the temporal cascade of KSHV lytic gene expression. Due to the location of the 18 S rRNA base m[7]G1639 in the ribosome, and movement with the mRNA and tRNAs, it has been suggested the base and modification could be involved in 40S subunit scanning and/or translocation[44]. Slower scanning of 40S pre-initiation complexes along mRNA 5'-UTRs has been reported to increase the translation of uORFs[45,46]. Therefore, BUD23 depletion and the reduction of methylation at G1639 may lead to slower scanning of the 40S pre-initiation complex and increased translation initiation of some uORF groups. An alternative mechanism could be similar to the CHOP uORF which causes ribosomes to stall, therefore reducing the

expression of the CHOP CDS[47]. The methylation of G1639 could therefore decrease the chance of a ribosome stalling at the uORF allowing more efficient translation of the downstream CDS. Either mechanism correlates with the increased association of ribosomes with some KSHV uORFs during KSHV lytic replication upon BUD23 depletion and reduced downstream CDS translation.

Our ribosome profiling data does not detect any significant changes in translational efficiency of human genes. We hypothesise this is because of the very low read depth of human genes in our dataset with <8% of the mRNA reads and <30% of the ribosome footprint reads mapping to human genes, leading to low confidence of changes in translational efficiency. However, the methylation of the 18S rRNA base G1639 must be required for the effective translation of a population of cellular transcripts. Baxter et al. implicated BUD23 and m[7]G1639 in the preferential translation of mRNAs with low 5'-UTR GC

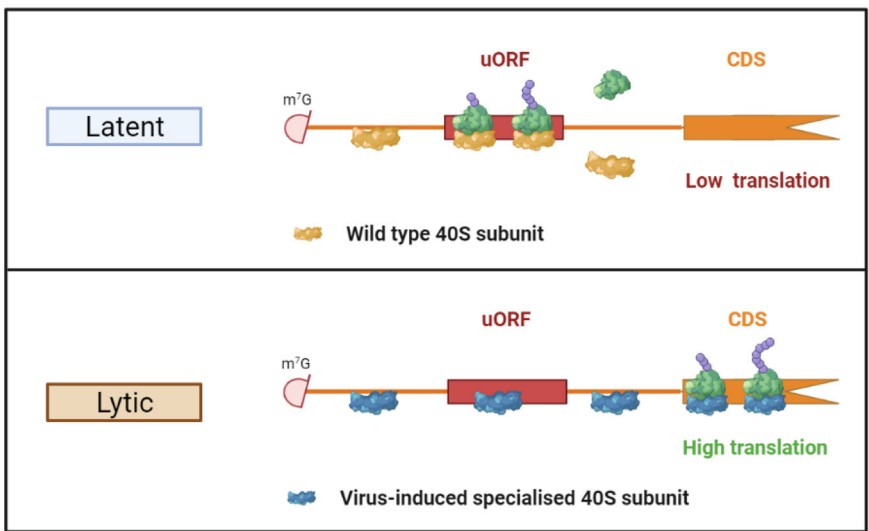

**Fig. 6 | KSHV co-opts ribosome biogenesis during lytic replication to produce virus-specific specialised ribosomes.** Enhanced association of BUD23, most likely driven by ORF11, with pre-40S ribosome complexes, during KSHV lytic replication drives the production of virus-specific specialised ribosomes. These ribosomes containing the 18 S rRNA m7G1639 modification reduce the translation of uORFs in late lytic KSHV mRNAs and consequently enhance the translation of the downstream CDS. Created using Biorender.

content[30]. Although, our data suggests that BUD23 could also be important for the effective translation of cellular transcripts which contain specific types of uORFs. As BUD23 impacts the translation of mRNAs with low 5'-UTR GC content and/or certain uORF types this could help explain the variety of symptoms observed in the genetic disease Williams syndrome in which BUD23 and other genes are deleted[31].

To summarise, KSHV manipulates ribosomes biogenesis through the viral protein ORF11 and increases the interaction of BUD23 with pre-40S ribosomal subunits. BUD23 and ORF11 regulate the translation of uORFs in late lytic KSHV mRNAs and the efficient translation of the downstream CDS, which is essential for the efficient production of new KSHV infectious virions. Overall, our data demonstrates a novel mechanism for the production of viral-induced specialised ribosomes which efficiently translate viral mRNAs for the production of new virions.

## Methods

### Reagents tables
Plasmids (Supplementary Table 1), primers and gRNAs (Supplementary Table 2) and primary antibodies (Supplementary Table 3).

### Cell culture
TREx BCBL1-Rta cells, a primary effusion lymphoma B cell line latently infected with KSHV and modified to contain doxycycline inducible myc-RTA, were a kind gift of Jae U. Jung (University of Southern California)[48]. TREx BCBL1-Rta cells were grown in RPMI1640 growth medium with glutamine (Gibco®), supplemented with 10% foetal bovine serum (FBS, Gibco®), 1% penicillin-streptomycin (Gibco®) and 100 mg/mL hygromycin B (Thermo Scientific). HEK-293T cells (American Type Culture Collection) were cultured in Dulbecco's modified Eagle's medium with glutamine (DMEM, Lonza) supplemented with 10% FBS (Gibco®) and 1% penicillin-streptomycin (Gibco®). All cell lines were tested negative for mycoplasma.

Cell lines were maintained by passaging every three to four days up to a passage number of 30. HEK-293T cells were passaged by mechanical disruption from the flask surface once reaching a confluency of 80%. The cells were diluted down in fresh DMEM growth media and reseeded at 10% confluency. TREx BCBL1-Rta cells were passaged upon reaching a density of $1.5 \times 10^6$ cells/ml by dilution with fresh RPMI growth media and reseeded at a density of $0.1 \times 10^6$ cells/ml. For virus reactivation, TREx BCBL1-Rta cells were induced using 2 mg/mL doxycycline hyclate (Sigma-Aldrich)[49].

### Lentivector expression and shRNA systems
Stable cell lines were generated in sub-confluent HEK-293T cells seeded were transfected with lentivirus packaging plasmids and an expression or shRNA plasmid[50,51]. The two packaging plasmids psPAX2 (6.5 µg/ml) and pVSV.G (6.5 µg/ml) and an expression or shRNA plasmid (12 µg/ml) were diluted in OPTI-MEM media containing lipofectamine 2000 (Invitrogen™) and co-transfected into HEK-293T cells. Six hours post-transfection the media was replaced with fresh DMEM without penicillin or streptomycin. At 48 h post-transfection, the lentiviral supernatants were collected and filtered through a 0.45 µm membrane (Merck Millipore). Then in the presence of polybrene (8 µg/ml) the filtered lentiviral supernatant was added to TREx BCBL1-Rta or fresh HEK-293T cells. TREx BCBL1-Rta cells were spin-inoculated (800 × g, 1 h, room temperature) then reseeded into six well plates. Six hours post-transduction the media was replaced with fresh RPMI or DMEM media. At 48 h post-transduction, the media was replaced with fresh selection RPMI or DMEM media containing puromycin (2 µg/ml, Sigma-Aldrich). The selection media was then changed, and the cells passaged every three to 4 days.

### CRISPR/Cas9 stable knockout cell lines
HEK-293T cells were transfected with the 3 plasmid lentiCRISPR v2 system. In 12-well plates, 4 µl of lipofectamine 2000 (Invitrogen™) was combined with 1 µg lentiCRISPR V2 plasmid expressing the guide RNA (gRNA) targeting the protein of interest, 0.65 µg of pVSV.G and 0.65 µg psPAX2. Two days post transfection the viral supernatant was harvested and filtered (0.45 µm pore, Merck Millipore) before transduction of TREx BCBL1-Rta cells in the presence of 8 µg/mL of polybrene (Merck Millipore). Virus supernatant was removed 6 h post transduction and fresh media added. Cells were maintained for 48 h before 2 µg/µl puromycin (Sigma-Aldrich) selection. Stable mixed population cell lines were maintained until confluency before single cell selection. Single cell populations were generated through serial dilution of ~50 cells distributed across a 96 well plate. Successfully cultured wells were maintained for 3–5 weeks with regular exchange of fresh media before transferal into 6 well plates. Upon confluency,

single cell clones were harvested and DNA extracted using Monarch Genomic DNA Purification Kit (New England Biolabs) as described in manufacturer's protocol. Extracted genomic DNA was subsequently sequenced (Source Bioscience, Sanger sequencing service) with ORF11 site specific primers. Editing efficacy and indels generated were assessed using TIDE (http://shinyapps.datacurators.nl/tide)[52].

## Affinity purification assays

For each Strep tag pulldown, TREx BCBL1-Rta cell lines expressing FLAG-2xStrep tagged ribosome biogenesis bait proteins and a control TREx BCBL1-Rta cell line were seeded at $0.75 \times 10^6$ cells/ml in 20 ml of RPMI selection or growth media. This was performed in duplicate and in half the flasks KSHV lytic replication was induced. After 24 h incubation the cells were harvested and cell pellets were lysed [10 mM Tris/HCl (pH 7.6), 100 mM KCl, 2 mM MgCl$^2$, 0.5% (v/v) NP-40, 1 mM DTT, 1% (v/v) Phosphatase inhibitor (v/v), and 1 × Roche protease inhibitor cocktail] for 20 min on ice. Magnetic Strep-Tactin®XT coated beads (IBA Lifesciences GmbH) were equilibrated by washing 40 µl of bead slurry three times in lysis buffer not containing phosphatase or protease inhibitor. The cell lysates were then clarified by centrifugation ($12,000 \times g$, 10 mins, 4 °C) and incubated with the equilibrated beads for 1 h on a rotator disk at 4 °C. The supernatant was then removed, and the beads washed four times in wash buffer I [10 mM Tris/HCl (pH 7.6), 100 mM KCl and 2 mM MgCl$_2$] with short a pulse vortex in between. The beads were then finally washed in wash buffer II [10 mM Tris/HCl (pH 7.6) and 2 mM MgCl$_2$] with a pulse vortex. The beads were either sent to the University of Bristol proteomics facility for tandem mass tagging (TMT) coupled to liquid chromatography (LC), mass spectrometry (MS) analysis, or total nucleic acid was extracted from the beads using TRIzol™ (Invitrogen™), as described by the manufacture, and run on a denaturing polyacrylamide gel or used for RT-qPCR, or the beads were diluted in Laemmli loading buffer before analysis by western blotting or silver stain.

For GFP-trap® and FLAG-trap affinity pulldowns, TREx BCBL1-Rta cell lines expressing, GFP, GFP-ORF11 or ORF11-FLAG were seeded at $0.75 \times 10^6$ cells/ml in 20 ml of RPMI selection or growth media. Alternatively, $1 \times 10^6$ HEK-293T cells were co-transfected with 1 µg ORF11-V5 and 1 µg of either GFP, GFP-importin α1 or GFP-importin α5 plasmid DNA using Lipofectamine® 2000 according to manufacturer's instructions (Invitrogen™)[53]. After 24 h, cells were lysed as described above and the same buffers used as above except using GFP-trap® or Fab-Trap™ affinity beads (ChromoTek™) according to manufacturer's instructions. Finally, the beads were either sent to the University of Bristol proteomics facility for TMT labelling coupled to LC/MS-MS analysis, or the beads were diluted in Laemmli loading buffer before analysis by western blotting.

## Quantitative proteomics

Strep-Tactin®XT and GFP-trap® beads bond with bait protein purified pre-ribosome complexes or GFP-ORF11 co-precipitated complexes, respectively, were sent in wash buffer II to the University of Bristol Proteomics facility for TMT coupled to LC-MS/MS. Briefly, samples were proteolytically digested and labelled with amine-specific isobaric tags yielding differentially labelled peptides of the same mass[54]. Labelled samples were then pooled and fractionated using Strong Anion eXchange chromatography before analysis by synchronous precursor selection MS3 on an Orbitrap Fusion Tribrid mass spectrometer (Thermo Fisher).

For each sample the background abundance values for each protein from the control TREx BCBL1-Rta cell lysate pulldown or control GFP only pulldown were taken away from the abundance value of each protein identified in the FLAG-2xStrep tagged ribosome biogenesis bait protein sample or GFP-ORF11 pulldown sample, respectively. For the FLAG-2xStrep tagged ribosome biogenesis bait protein samples a cutoff abundance value of 150 was then selected and all

proteins with a lower value were discarded from further analysis. For GFP-ORF11 pulldown samples a cutoff abundance of >1% of each protein compared to the GFP-ORF11 bait and >1.5 fold enrichment over the control GFP only pulldown. All processed mass spectrometry data is provided in Supplementary Data 1 and 2.

## Silver stain analysis

Isolated pre-40S ribosome complexes were run on 12% polyacrylamide gels [12% (v/v) acrylamide/bis-acrylamide 37.5:1, 375 mM Tris/HCl (pH 8.8), 0.1% (w/v) SDS, 0.12% (w/v) APS, 0.012% TEMED (v/v)] and a stacking region [5% (v/v) acrylamide/bis-acrylamide 37.5:1, 125 mM Tris/HCl (pH 6.8), 0.1% (w/v) SDS, 0.08% (v/v) APS, 0.008% (v/v) TEMED]. Gels were resolved using a Mini-PROTEAN gel electrophoresis system (Biorad) in running Buffer [25 mM Tris/HCl, 192 mM Glycine and 0.1 % (w/v) SDS]. Gels were first fixed in 50% (v/v) Ethanol and 10% (v/v) Acetic Acid, then 5% (v/v) Ethanol and 1% (v/v) Acetic Acid for 30 min and 15 min respectively on a shaker at room temperature. The gels were then washed three times in ultrapure water for 5 min. Next, the gels were sensitised in 0.02% (w/v) Sodium Thiosulphate for 2 min and then washed again three times in ultrapure water (Milli-Q® Advantage A10 Water Purification System, Merck) for 30 s. The gels were stained in 0.1% (w/v) Silver Nitrate and 0.08% (v/v) formaldehyde in the dark for 20 min and washed again three times in ultrapure water for 20 seconds. The gels were finally developed in 2% (w/v) Sodium Carbonate, 0.04% (v/v) Formaldehyde and 0.0004% (w/v) Sodium Thiosulphate until protein bands were visible, then the reaction stopped using 5% (v/v) Acetic Acid. A photograph of the gel was then taken using a G:Box Chemi XX9 imager.

## Immunoblotting

Protein samples were run on 12–15% polyacrylamide gels and transferred to nitrocellulose Hybond™-C (GE Healthcare) membranes via Bio-Rad Trans-blot Turbo transfer machine[55,56]. Membranes were blocked with TBS + 0.1% v/v Tween® 20 and 5% w/v dried skimmed milk powder. Membranes were probed with relevant primary and secondary HRP-conjugated IgG antibodies, treated with ECL Western Blotting Substrate (Promega), and detected using a G:Box Chemi XX9 imager (Alpha Metrix).

## Denaturing polyacrylamide gel electrophoresis

Total nucleic acid, isolated from FLAG-2×Strep-PNO1 affinity pulldowns was resuspended in formamide loading dye [95% (v/v) Deionised Formamide, 20 mM EDTA, 0.05% (w/v) Xylene Cyanol, 0.05% (w/v) Bromophenol Blue] and boiled for 5 min at 95 °C. The samples were then loaded onto a 8% denaturing urea polyacrylamide gel [50% (w/v) urea, 1 × TBE Buffer, 8% Acrylamide/Bis-acrylamide 19:1, 0.08% (v/v) APS, 0.008% (v/v) TEMED] alongside a 1 Kb Plus DNA Ladder (Thermo Fisher) and resolved at 200 V for 2 h in TBE buffer. The gel was then stained for 20 min in the dark with 1:10,000 SYBR™ Gold in TBE buffer and imaged using a G:Box Chemi XX9 imager.

## Two-step reverse transcription quantitative PCR (RT-qPCR)

Total RNA was isolated from cell pellets using Monarch® Total RNA Miniprep Kits (New England Biolabs), as described by the manufacturer. Purified RNA was quantified by ultraviolet spectrophotometry and 1 µg used for reverse transcription by LunaScript® RT SuperMix Kit (New England Biolabs), as described by the manufacturer. Quantification of cDNA or DNA was determined by qPCR using a Rotor-Gene Q platform (QIAGEN) with sequence specific primers for each gene. Samples were analysed in technical duplicates and three or more biological repeats. Amplification was performed in 20 µl reaction volumes with 40 ng template cDNA using GoTaq® qPCR master mix (Promega) according to the manufacturer's instructions, with a standard three-step melt program (95 °C for 15 s, 60 °C for 30 s, 72 °C for 20 s). To assess primer amplification efficiency (AE) for each

gene of interest a standard curve was constructed using a pool of cDNA derived from unreactivated or reactivated cells. Five different dilutions of pool cDNA were quantified to generate a standard curve. The slope of the standard curve was used to calculate the AE of the primers using the formula: $AE = (10^{-1/slope})$. For gene expression analysis all genes of interest were normalised against the housekeeping gene GAPDH ($\Delta C_T$), and to a reference sample ($\Delta\Delta C_T$).

## Global translation assay

TREx BCBL1-Rta cells stably expressing scrambled shRNA or shRNAs targeting BUD23 were seeded into a 6-well plate at $1 \times 10^6$ cells/well with 2 ml RPMI selection media without methionine which was chased out over 1 h. Cells were treated with Click-iT® L-azidohomoalanine (AHA, 40 μM, Thermo Fischer) for 3 h, then washed in PBS and fixed in PBS containing 4% (v/v) paraformaldehyde for 15 min. Cells were again washed in PBS and permeabilised using PBS containing 0.25% Triton X-100 for 15 min. AHA was stained with a Click-iT® reaction kit (Thermo Fischer) using an alkyne Alexa Fluor™ 488, 1 μM, as described by the manufacturer. Finally, cells were washed in PBS containing 1% BSA and resuspened in PBS containing 0.5% BSA before fluorescence quantification by CytoFlex S Benchtop Flow Cytometer (Beckman Coulter).

## Cell proliferation assay

TREx BCBL1-Rta or HEK-293T cells stably expressing scrambled shRNA, shRNAs targeting BUD23 or overexpressing ORF11-FLAG were seeded into a 6-well plate at $0.2 \times 10^6$ cells/well with 2 ml fresh RPMI or DMEM selection media, respectively. The cells were grown for 48 h and the counted at each 24 h interval.

## 18 S rRNA 1639 m$^7$G methylation quantification

Total RNA was isolated from latent or 24 h, 36 h or 48 h post lytic reactivation TREx BCBL1-Rta cells expressing scrambled shRNA or shRNAs targeting BUD23 using TRIzol™ (Thermo Fischer) as described by the manufacturer. Purified nucleic acid pellets were resuspended in 1.0 M Tris-HCl (pH 8.2) and first reduced, specifically at m$^7$G sites, with 0.2 M NaBH$_4$ for 30 min on ice in the dark, as previously described[57]. The reaction was stopped with a sodium acetate, isopropanol precipitation of the RNA over night at −80 °C. The RNA was pelleted by centrifugation (16,000 × g, 15 min, 4 °C) and washed in ethanol twice before being resuspended in 1.0 M Tris-HCl (pH 8.2). Reduced m$^7$G sites were then cleaved by β-elimination using 1 M aniline/acetate (pH 4.5) in the presence of 50 μg m$^7$GTP carrier on ice in the dark for 15 min. The reaction was again stopped by precipitation of the RNA but for 2 h at −80 °C and the RNA purified as described above. The purified RNA was resuspended in Nuclease-Free Water then reverse transcribed, as described above, and qPCR analysis performed with primers specific for total 18S rRNA and primers flanking the 18S rRNA 1639 m$^7$G cleavage site. The comparative ΔΔCT was performed firstly between total 18S and the 18S 1639 site then between the control and experimental conditions.

## Viral genome copies and virus reinfection assays

TREx BCBL1-Rta cells expressing Scr shRNA, BUD23 specific shRNAs or Scr gRNA or ORF11 gRNAs were seeded into a 6-well plate at $0.75 \times 10^6$ cells/well with 2 ml fresh RPMI growth media (without puromycin)[58]. KSHV reactivation was induced for 72 h and the culture medium, containing released virions, was then centrifuged to remove cells and debris and mix at a 1:1 ratio with fresh complete DMEM growth media which was incubated for 48 h with naive HEK-293T cells at 40% confluency. Total DNA was harvested from TREx BCBL1-Rta cell pellets, using Monarch® Genomic DNA Purification Kits (New England Biolabs), and viral genome copies quantified by qPCR of the viral gene ORF57. Total RNA was harvested from HEK-293T cells and RT-qPCR analysis performed for the viral mRNA ORF57 to assess efficiency of virus reinfection.

## Immunofluorescence

HEK-293T cells or TREx BCBL1-Rta cells expressing ORF11-FLAG were grown on sterilised glass coverslips treated with Poly-L-Lysine (Sigma-Aldrich). After 24 h cells were washed in PBS and fixed in PBS containing 4% (v/v) paraformaldehyde for 10 min, washed twice in PBS and permeabilised using PBS containing 1% Triton X-100 for 10 min[58,59]. Coverslips were then incubated with anti-FLAG or anti-LANA and Alexa Flour 546 (Invitrogen), goat anti-Rat IgG or goat anti-Rabbit IgG respectively, for 1 h each at 37 °C before being mounted onto microscope slides using Vectashield® with DAPI. Slides were visualised on a Zeiss LSM 700 laser scanning confocal microscope and images analysed using Zen® blue 2.0 (Zeiss).

## Polysome profiling

TREx BCBL1-Rta cells expressing Scr shRNA, BUD23 specific shRNAs or ORF11-GFP were treated with cycloheximide (Sigma) at 100 μg/ml for 3 min at 37 °C. A total of ~$50 \times 10^6$ cells were pelleted and washed (1 x PBS, 100 μg/ml cycloheximide) then lysed in ice cold buffer [50 mM Tris-HCl pH8, 150 mM NaCl, 10 mM MgCl$_2$, 1 mM DTT, 1% IGEPAL, 100 μg/ml cycloheximide, Turbo DNase 24 U/mL (Invitrogen), RNasin Plus RNase Inhibitor 90 U (Promega), 1 × protease inhibitor cocktail (Roche)] for 45 min. Lysates were clarified by centrifugation (12,000 × g, 10 min, 4 °C) and the resulting supernatants applied to continuous sucrose gradients of either 5–45% or GFP-ORF11 lysates applied to 5–30% (10 mM MgCl$_2$, 50 mM Tris/HCl (pH 7.6), 150 mM NaCl, 1 mM DTT, 100 μg/ml cycloheximide and 1× protease inhibitor cocktail). Gradients were then subjected to ultracentrifugation (121,355 × g 3.5 h, 4 °C) in SW-40 rotor. Fractions from each gradient were collected using a Gradient Fractionator (BioComp) and the RNA profile was measured by absorbance (254 nm) across the gradient in real time using an EM-1 Econo UV Monitor (Bio-Rad).

## Ribosome profiling

KSHV lytic replication was induced in TREx BCBL1-Rta cells expressing a Scr shRNA or shRNA targeting BUD23 for 36 h. Cells were processed and polysome profile analysis performed as described above, in addition 20% of clarified cell lysates were saved as a total mRNA input for downstream processing and analysis. Total nucleic acid was extracted from input samples using TRIzol™ LS (Invitrogen™), as described by the manufacturer. Sucrose fractions containing 80S and polysome ribosomes were used for poly-ribo-seq[60,61]. Fractions for each sample were pooled and diluted (100 mM Tris/HCl (pH 7.6), 30 mM NaCl and 10 mM MgCl$_2$) to a 10% sucrose concentration and ribosomes footprinted with 1 U/1×10$^6$ cells RNase I (Thermo Fisher), rotating overnight at 4 °C. The reaction was stopped with Super-RNaseIN (200U/gradient, Thermo Fisher), rotating for 5 min at room temperature. Sucrose samples were concentrated with 30 kDa cut off filter units (Millipore) and loaded onto 5–45% sucrose gradients. Gradients were processed as described above and just the 80S fraction collected containing footprinted ribosomes and the RNA precipitated 1:1 isopropanol, 300 mM NaCl and glycoblue, overnight at −80 °C. Fractions for each were pooled through resuspension in 100 μl Nuclease-Free Water, and the same done for the mRNA input samples. Samples were treated with TURBO DNase (Thermo Fisher) according to manufacturer's instructions and the RNA purified with TRIzol™ LS (Invitrogen™), as described by the manufacture. PolyA RNA was selected from input mRNA samples using a Dynabeads™ mRNA DIRECT™ Purification Kit (Invitrogen™), according to manufacturer's instructions and the RNA fragmented in 2 mM EDTA, 10 mM Na$_2$CO$_3$ and 90 mM NaHCO$_3$ for 20 min at 95 °C. 28–34 nt ribosome footprints and 50–80 nt polyA RNA samples were gel purified in 10% (w/v) polyacrylamide-TBE-urea gel at 300 V for 3.5 h in 1X TBE. Fragment ends were repaired using T4 PNK treatment (New England Biolabs) as described by the manufacturer. Ribosome footprints were subjected to rRNA depletion using a NEBNext® rRNA depletion kit (New England

Biolabs) according to the manufacturers protocol. The NEBNext® Multiplex Small RNA Library Prep Set for Illumina® (New England Biolabs) was used to generate the libraries for next generation sequencing. Resulting cDNA was PCR amplified and gel purified prior to sequencing. Libraries were subjected to 75 bp single end RNA Seq using NovaSeq 6000 Illumina sequencer (Novogene).

Raw sequencing data was demultiplexed and single-end sequencing quality control was assessed through FastQC (www.bioinformatics. babraham.ac.uk/projects/fastqc). An average of 128 million reads were sequenced per sample. Both adaptors and low-quality bases (QV < 20) were trimmed from reads' extremities using Cutadapt (v3.2) with minimum read length of 25 bp, and untrimmed outputs discarded for Ribo-Seq reads[62]. All libraries were mapped against Human hg38 rRNAs (Gencode v36) (https://www.gencodegenes.org/human/release_36. html) and tRNA sequences (GtRNAdb 18.1) (http://gtrnadb.ucsc.edu/ archives.html) using Bowtie2 v.2.3.4.2 (--sensitive-local -N1 -k1), and then removed using Samtools v1.9 with the -f 4 option[63,64]. STAR aligner with default parameters was used for alignments of each QC-processed library against both Human (Gencode v36 - GRCh38p13 primary assembly) and KSHV (NCBI - GQ994935.1) genomes, separately[65]. STAR-generated BAM output files were used for assigning read counts to CDS features in each genome with featureCounts, disregarding multi-mapper reads (not invoking -M option) and assigning reads to all overlapped features (invoking -O option)[66]. Gencode v36 primary assembly gtf annotation file was used for Human counts, whereas KSHV 2.0 annotation for virus[16]. Read counts' tables generated by featureCounts for each organism were then used as input for differential translation (DT) analyses with RiboRex relying on the DeSeq2 negative binomial distribution model and a 0.05 FDR threshold[67,68]. Both genome-aligned read count assessments were submitted to a multi-dimensional scaling analysis using the plotMDS function from the EdgeR package[69]. Those tools were run under the R environment version 4.0.4.

### Dual luciferase reporter assay

Various KSHV mRNA 5'-UTRs containing uORFs were cloned upstream of the Renilla luciferase gene in a psiCHECK™-2 vector (Promega) using a Gibson Assembly® Cloning Kit (New England Biolabs). As control plasmids, the start codon of each uORF was mutated using a Q5® Site-Directed Mutagenesis Kit (New England Biolabs). The sequence additions and mutations of all plasmids were confirmed by DNA sequencing (Source Bioscience).

Luciferase activity was detected using the Dual-Luciferase Reporter Assay System (Promega)[70]. HEK-293T cells expressing either a non-targeting scrambled shRNA, shRNAs targeting BUD23 or overexpressing ORF11-FLAG were seeded in triplicate in 12-well culture plates (Thermo Fisher) at a density of $3 \times 10^5$ cells per well and grown for 24 h. Cells were transfected with the respective plasmids using Lipofectamine® 2000 according to manufacturer's instructions (Invitrogen™) and incubated for a further 24 h. Media was removed from the culture wells and cells washed gently with 100 µl PBS. 100 µl 1x passive lysis buffer was added to the cell monolayer which was rocked for 15 min and then 10 µl of each lysate was transferred to tissue culture treated white microplates (Greiner Bio-One). Luciferase measurements were carried out in a FLUOstar Optima microplate reader (BMG Labtech Ltd), with injectors 1 and 2 being used to dispense 50 µl of Luciferase Assay Reagent II and Stop & Glo Reagent respectively. Renilla luciferase activity was normalised to Firefly luciferase activity.

### Statistical analysis

Except otherwise stated, graphical data shown represent mean ± standard deviation of mean (SD) using three or more biologically independent experiments. Differences between means was analysed by unpaired Student's t test or two-tailed ANOVA test calculated using Graphpad Prism 9 calculator. Statistics was considered significant at $p < 0.05$, with $*P < 0.05$, $**P < 0.01$, $***P < 0.001$.

### Reporting summary

Further information on research design is available in the Nature Portfolio Reporting Summary linked to this article.

## Data availability

Quantitative mass spectrometry datasets have been deposited to the PRIDE, Proteomics Identifications Database and are publically available under project accession codes PXD032318 for Pre-40S affinity purification proteomics and PXD032367 for KSHV ORF11 affinity purification proteomics. The Poly-ribo-seq dataset has been deposited to NCBI GEO, Gene Expression Omnibus and is publically available under GEO accession code GSE199095. Other publically available datasets used in this analysis include Human hg38 rRNAs (Gencode v36) (https://www.gencodegenes.org/human/release_36.html) and tRNA sequences (GtRNAdb 18.1) (http://gtrnadb.ucsc.edu/archives.html). The reporting summary for this article is available in the Supplementary Information section. All the other data supporting this study are available within this Article, Supplementary Information, and Source Data. Source data are provided with this paper.

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

## Acknowledgements

We are indebted to Professor Jae Jung (UCLA) for the TREx BCBL1-Rta cell line, Dr. Edwin Chen (University of Westminster) for the psPAX2 and pVSV.G plasmids, Prof. Yan Yuan (University of Pennsylvania) for the ORF11-V5 plasmid and Prof. Britt Glaunsinger (University of California, Berkeley) for the ORF59 antibody. We also thank Dr. Kate Heesom (Proteomics Facility, University of Bristol, UK) for proteomic technical assistance and initial sorting of proteomic data. Figure 6 was created using Biorender. This work was supported by Wellcome Trust to J.C.M. and A.W. (203826/Z/16/Z), BBSRC to A.W. and J.L.A. (BB/N014405/1) and a University of Leeds Mary & Alice Smith Endowed Research Scholarship to E.M.H. and A.W.

## Author contributions

A.W. and J.L.A. conceived the study and acquired project funding. J.C.M., E.M.H., S.S., T.J.M., K.L.H. performed the experiments and J.C.M., E.M.H., S.S., T.J.M., K.L.H., J.L.A., A.W. analysed the resulting data. EJRV analysed Ribosome sequencing datasets. The original draft was written by J.C.M., E.M.H., S.S., A.W., and all authors reviewed and edited the final version.

## Competing interests

The authors declare no competing interests.
