## [Peer Review File · Nature Communications]

Kaposi's sarcoma-associated herpesvirus induces specialised ribosomes to efficiently translate viral lytic mRNAsReviewers' Comments:

Reviewer #1:

Remarks to the Author:

In this very interesting manuscript, A. Whitehouse's group shows that during the KSHV lytic cycle specialized ribosomes are assembled to permit the translation of late viral proteins and the production of virions. This represents an innovative observation, which should be of interest to a broader community of virologists and life scientists. The experimental data shown in the manuscript are of high technical quality, convincing and support the authors' conclusions. I only have two experimental suggestions and one minor comment:

1. Lines 191/192: Based on the result shown in Figure 4a-f and 4g, the authors conclude that "N7-methylation of G1639 may (space) be required for the effective translation of late KSHV genes or genes involved in late gene expression". To corroborate that this increased N7-methylation of 18S RNA shown in figure 4g is really due to BUD23, the authors should carry out a similar time course of 18S RNA methylation in their BUD23 knockdown cells shown in figure 4 a-e.

2. Line 249: In figure 5e the authors show that cells harboring KSHV genome with a mutated ORF11 produce fewer infectious virus particles than those containing a KSHV wt genome. The authors should show that the lytic cycle in the ORF11 Stop cells arrests before the late phase of lytic replication, i.e. that the expression of early proteins is still possible in the ORF11 stop cells but the expression of late proteins is not.

Minor point:

Lines 175-179: the reference to panels b-e of figure 4 in the text is wrong: panels b and c show the effect of BUD23 knockdown on K8.1 and ORF65 mRNA, while panels d and e refer to protein levels - in the text it is the opposite.

Reviewer #2:

Remarks to the Author:

In this manuscript by Murphy et al., entitled "Kaposi's sarcoma-associated herpesvirus induces specialised ribosomes to efficiently translate viral lytic mRNAs", the authors propose an alternative translation control mechanism during KSHV lytic infection via the usurpation of specialized ribosomes. The authors show that during lytic KSHV infection, there is an enhanced association between the viral protein ORF11 and the ribosomal biogenesis proteins BUD23 and NOC4L. This appears to be essential during the viral replication cycle as BUD23 depletion is shown to affect protein levels of KSHV late genes and overall infectious particles. Finally, the authors did ribosome profiling in BUD23 depleted cells and suggest that the depletion of BUD23 increases translation of uORF, and consequently decreases translation of main CDS, for specific KSHV genes. The data presented in the manuscript support a potential effect of BUD23 on KSHV replication and do point to a novel mechanism by which the virus could usurp the translation control machinery to facilitate translation of late viral mRNAs during the lytic cycle.

The paper is well written and reports on very interesting observations. However, the data in their present form are insufficient to conclusively establish BUD23 as a translation modifier for KSHV transcripts via specialised ribosomes that bypass certain viral uORFs. The mechanism of ORF11 modulation on BUD23 function and ribosome biogenesis/assembly is also not explored. Altogether, the manuscript needs significant additional experiments to qualify for publication in Nature Communications.

The major concerns and questions are listed below:

Figure 2 : To demonstrate that no differences exist in global translation between Scrambled and

BUD23-shRNA cells, the authors should consider puromycin or 35S labeling assays.

Figures 3 and 4 could be combined into one figure, as they convey the same message that late (but not early) genes are affected by BUD23 knockdown. There is still a significant reduction in mRNA for K8.1 and ORF65 in the KD cells, so it could be seen as premature to suggest that BUD23 depletion leads to only a decrease in translation of these viral genes. The author should consider demonstrating by polysome profiling that the distribution of the mRNAs encoding for K8.1 and ORF65 is affected in BUD23 KD cells compared to Sc cells. This would also address concerns that these viral genes don't appear to be under the same control when assessed by ribosome profiling or the luciferase assay presented later in the manuscript.

In figure 4F (and also 5E), the authors report their viral re-infection assay as a fold change relative to scrambled, and show a ~90% reduction. This is performed by RT-qPCR. How does this compare to overall re-infection titers between Sc and KD cells? In essence, is the reported reduction less than a log when measured by PCR and normalized to control? The authors should report both methods.

Figure 5: Based on the distribution of ORF11 and BUD23 on the polysome fractionation, BUD23 doesn't appear to be primarily associated with ORF11 in the pre-40S gradient, perhaps only in fractions 1-2. A pulldown of ORF11 or BUD23 in those fractions could confirm better the association between two proteins? The authors state that "ORF11 specifically interacts with pre-ribosomal and ribosomal proteins of the 40S subunit". Is this interaction direct or through the entire small ribosomal complex? If the samples are treated with RNase, would there still be interactions between ORF11 and BUD23? Can the authors also probe for large ribosomal proteins to demonstrate specificity? It would be also important for the authors to improve upon their immunofluorescence data to more clearly demonstrate the colocalization of ORF11 with sites of the nucleolus

Figure 6: There is a need for additional data to confirm the effect on translation control. First, what is the translation efficiency and RPF density of ORF65, shown in Figure 4c,e, to be affected at the translation level? Are TE and RPF density also similarly reduced on the CDS of ORF65? The authors suggest that the effect of BUD23 knockdown is an increase in uORFs translation and consequently a reduction in translation of the main CDS. The data shown in Figure 6b is insufficient to support this notion and also kind of counterintuitive. An increase in uORF of ORF69, 75, and 61 should decrease the directly related CDS of ORF69, 75, and 61. Similarly, a decrease in translation efficiency of the main CDS of ORF28 should correlate with an increase in translation efficiency of the associated uORF. Mapping of the ribosomal reads could provide a visual information on this. The authors need to show those data to support a dysregulated KHSV transcript control that includes an increased translation of uORF and a decreased translation of main CDS caused by BUD23 depletion. Could the author provide/quantify the change in TE of uORFs and the corresponding ORFs of all KSHV genes in BUD23 depleted cells?

In that respect, what is the effect of BUD23 depletion during KHSV infection on translation of host transcripts and uORF-containing host transcripts? Can the authors look into this within their ribosome profiling data? This would be an important aspect to discuss as well.

Figure 6e,f,g: Why did the author compare the shRNA1 and shRNA2 to ORF11 overexpressed cells? The luciferase expression level of shRNA 1, shRNA 2, and ORF11 expressed cells should be compared to shSCR control. There doesn't seem to be a significant effect of BUD23 depletion by shRNA 1 and shRNA 2 on luciferase translation under the 5'UTR of ORF28 and ORF34. Also the ribosome profiling data suggested that the uORF in the 5'UTR of ORF69 is induced translationally in BUD23 depleted cells, can the author validate this observation using a similar reporter assay: whether to fuse luciferase CDS right after the uORF of ORF69, or adding a small tag (HA, FLAG, etc...) fused to the 3'end of this uORF, so that uORF translation can be assessed by Western blotting.

Does the 5'UTR of ORF69, ORF28 and ORF34 alter transcription levels also? The authors should measure luciferase mRNA level (by RT-qPCR) in the experiments in panel e,f,g to rule out the

transcriptional effect of the 5'UTRs.

Additional question, what is the effect of ORF11 association with BUD23 on ribosome biogenesis? How does this association modify the 40S subunit, or the fully assembled ribosome, to decrease recognition and translation of KHSV uORFs/CDS?

Reviewer #3:

Remarks to the Author:

This is an innovative paper in which the authors take an affinity-tag proteomic approach to determine whether KSHV infection manipulates ribosome biogenesis during its reactivation from latent to lytic phase infection.

They make a number of significant findings:

1 They find enhanced association of BUD23 with small ribosomal subunit precursor complexes during lytic KSHV infection. This appears to be critical for viral gene expression and loss of BUD23 leads to the reduced production of infectious virions.

2 They find that the previously uncharacterized KSHV lytic protein ORF11 associates with the pre-40S ribosomal subunit precursor complexes and is required for the production of new infectious virion. In the absence of ORF11 (mutated ORF11) they find reduced production of infectious virus.

This was an interesting, original and well-written manuscript. The experiments appeared to be rigorously performed and in general the data were convincing. I would be grateful if the authors could add molecular weight markers to all their protein gels to make them interpretable.

Is it feasible to perform a more subtle experiment with the ORF11 mutated virus? Does complementation with exogenous ORF11 rescue the ORF11 mutated virus. And if so – are they able to identify point mutation which prevent ORF11 association with the pre-40S complex?

REVIEWER COMMENTS

We thank the reviewers for their positive comments to which we provide a point-by-point response below. Changes are highlighted in the marked up version of the revised manuscript. Please note line/page numbers correspond to the marked up version of the modified manuscript.

Reviewer #1 (Remarks to the Author):

In this very interesting manuscript, A. Whitehouse's group shows that during the KSHV lytic cycle specialized ribosomes are assembled to permit the translation of late viral proteins and the production of virions. This represents an innovative observation, which should be of interest to a broader community of virologists and life scientists. The experimental data shown in the manuscript are of high technical quality, convincing and support the authors' conclusions. I only have two experimental suggestions and one minor comment:

1. Lines 191/192: Based on the result shown in Figure 4a-f and 4g, the authors conclude that "N7-methylation of G1639 may (space) be required for the effective translation of late KSHV genes or genes involved in late gene expression". To corroborate that this increased N7-methylation of 18S RNA shown in figure 4g is really due to BUD23, the authors should carry out a similar time course of 18S RNA methylation in their BUD23 knockdown cells shown in figure 4 a-e.

We have performed this experiment and results show that during KSHV lytic replication the N⁷-methylation of the 18S rRNA base G1639 remains significantly reduced when BUD23 is depleted compared to scrambled control cells. Lines 215-216 and Extended Data Fig. 11.

2. Line 249: In figure 5e the authors show that cells harboring KSHV genome with a mutated ORF11 produce fewer infectious virus particles than those containing a KSHV wt genome. The authors should show that the lytic cycle in the ORF11 Stop cells arrests before the late phase of lytic replication, i.e. that the expression of early proteins is still possible in the ORF11 stop cells but the expression of late proteins is not.

We have quantified the levels of early and late protein production in ORF11 deficient cells post reactivation. As expected late protein levels (e.g. ORF65) were significantly reduced in the ORF11 deficient cells. In contrast, there was only a small reduction in early protein levels (e.g. ORF57), showing that early protein expression is still possible in ORF11 deficient cells. In addition, an ORF11 overexpression construct was stably transduced into a CRISPR ORF11 deficient cell line to demonstrate that ORF11 reintroduction can rescue late ORF 65 protein levels. The new data is described in Lines 36-368 and presented in Fig 4f.

Minor point:

Lines 175-179: the reference to panels b-e of figure 4 in the text is wrong: panels b and c show the effect of BUD23 knockdown on K8.1 and ORF65 mRNA, while panels d and e refer to protein levels - in the text it is the opposite.

Figure 4 has now been edited and combined with figure 3 along with the K8.1 and ORF65 mRNA quantification been placed in extended data fig. 8. All references have been updated and corrected in the text.

Reviewer #2 (Remarks to the Author):

In this manuscript by Murphy et al., entitled “Kaposi’s sarcoma-associated herpesvirus induces specialised ribosomes to efficiently translate viral lytic mRNAs”, the authors propose an alternative translation control mechanism during KSHV lytic infection via the usurpation of specialized ribosomes. The authors show that during lytic KSHV infection, there is an enhanced association between the viral protein ORF11 and the ribosomal biogenesis proteins BUD23 and NOC4L. This appears to be essential during the viral replication cycle as BUD23 depletion is shown to affect protein levels of KSHV late genes and overall infectious particles. Finally, the authors did ribosome profiling in BUD23 depleted cells and suggest that the depletion of BUD23 increases translation of uORF, and consequently decreases translation of main CDS, for specific KSHV genes. The data presented in the manuscript support a potential effect of BUD23 on KSHV replication and do point to a novel mechanism by which the virus could usurp the translation control machinery to facilitate translation of late viral mRNAs during the lytic cycle.

The paper is well written and reports on very interesting observations. However, the data in their present form are insufficient to conclusively establish BUD23 as a translation modifier for KSHV transcripts via specialised ribosomes that bypass certain viral uORFs. The mechanism of ORF11 modulation on BUD23 function and ribosome biogenesis/assembly is also not explored. Altogether, the manuscript needs significant additional experiments to qualify for publication in Nature Communications.

The major concerns and questions are listed below:

Figure 2 : To demonstrate that no differences exist in global translation between Scrambled and BUD23-shRNA cells, the authors should consider puromycin or 35S labeling assays.

Global translation was assessed by incorporation of Click-IT™ L-Azidohomoalanine (Thermo Fisher) and detected by labelling with an alkyne Alexa Fluor™ 488 which was quantified using flow cytometry. Results show no change in global translation between Scrambled and BUD23-shRNA cells (see Lines 162-163 and Fig. 2f).

Figures 3 and 4 could be combined into one figure, as they convey the same message that late (but not early) genes are affected by BUD23 knockdown. There is still a significant reduction in mRNA for K8.1 and ORF65 in the KD cells, so it could be seen as premature to suggest that BUD23 depletion leads to only a decrease in translation of these viral genes. The author should consider demonstrating by polysome profiling that the distribution of the mRNAs encoding for K8.1 and ORF65 is affected in BUD23 KD cells compared to Sc cells. This would also address concerns that these viral genes don’t appear to be under the same control when assessed by ribosome profiling or the luciferase assay presented later in the manuscript.

As suggested, Figures 3 and 4 have been combined and the mRNA quantification of the viral genes, ORF57, ORF59, K8.1 and ORF65 has been moved to extended figure 8. Quantification of actively translating K8.1 and ORF65 mRNAs isolated from polysome profiles of BUD23 depleted cells compared to scrambled control cells show a significant reduction, lines 178-181 and Extended Data Figure 9. The effect of BUD23 depletion on KSHV lytic replication causes a breakdown of the late lytic gene cascade. We agree that this breakdown does cause a reduction in transcription of K8.1 and ORF65 however the larger effect we observe is the drop in translation of these late lytic proteins when BUD23 is depleted. We have therefore modified the conclusions from these data to reflect both the collapse of the KSHV lytic cascade and consequent reduction in transcription but also the larger drop in translation that is observed when BUD23 is depleted, lines 199-209.

In figure 4F (and also 5E), the authors report their viral re-infection assay as a fold change relative to scrambled, and show a ~90% reduction. This is performed by RT-qPCR. How does this compare to overall re-infection titers between Sc and KD cells? In essence, is the reported reduction less than a log when measured by PCR and normalized to control? The authors should report both methods.

To assess viral titre during the re-infection assay we would have ideally liked to perform a plaque assay. Unfortunately, KSHV does not lyse monolayer cultures including the HEK-293T cells that were used for viral re-infection. To confirm our RT-qPCR analysis, we have also fluorescently stained re-infected HEK-293T cells with the viral marker LANA and observed re-infected cells by confocal microscopy. These results further demonstrate the almost total loss of viral infectious virions from viral supernatants produced from BUD23 depleted cells compared to scrambled control cells (Lines 204-207 and extended data fig. 10). Similar results were observed for viral re-infection assays in HEK-293T cells stained for LANA with viral supernatants produced from cells infected with KSHV ORF11 knockout compared to wild type KSHV control cells (Line 369 and extended data fig. 15).

Figure 5: Based on the distribution of ORF11 and BUD23 on the polysome fractionation, BUD23 doesn't appear to be primarily associated with ORF11 in the pre-40S gradient, perhaps only in fractions 1-2. A pulldown of ORF11 or BUD23 in those fractions could confirm better the association between two proteins? The authors state that "ORF11 specifically interacts with pre-ribosomal and ribosomal proteins of the 40S subunit". Is this interaction direct or through the entire small ribosomal complex? If the samples are treated with RNase, would there still be interactions between ORF11 and BUD23? Can the authors also probe for large ribosomal proteins to demonstrate specificity? It would be also important for the authors to improve upon their immunofluorescence data to more clearly demonstrate the colocalization of ORF11 with sites of the nucleolus

We have attempted to perform an immunoprecipitation between ORF11 and Bud23 after polysome profiling, however this was unsuccessful. However we believe this is a technical issue association with the ribosome isolation protocol which leads to the disruption of protein-protein interactions, as we could not precipitation known interactions between ribosomal proteins as positive controls (data not shown). As previously shown in Figure 4a, we demonstrate that ORF11 interacts with pre-ribosomal subunits, therefore we have modified the text to state that at present which pre-ribosomal proteins ORF11 interacts with directly is unknown (lines 290-291).

Pulldowns of ORF11 treated with RNase have been included (Extended Data Fig. 12 and Lines 289-290) which show the interaction between ORF11 and BUD23 and eS19, a 40S subunit ribosome protein, are rRNA-dependent.

Probing ORF11 pulldowns for the large ribosomal subunit protein uL23 (Fig. 4a) demonstrates a potential interaction with the large ribosomal subunit. From our data ORF11 appears to associate with pre-40S ribosomal complexes until very late stages in their biogenesis and potentially still with free mature 40S complexes. While the late nucleolus and nucleoplasmic stages of 40S and 60S ribosomal subunit biogenesis are distinct, the cytoplasmic stages are much more intimately intertwined, including quality control docking of mature 60S subunits with pre-40S subunits, discussed in lines 470-476. Therefore we hypothesise that this is why we see some interaction of ORF11 with the large ribosomal subunit protein uL23. Furthermore, we have removed the word 'specifically' from line 230 to prevent any ambiguity around this data.

Further immunofluorescence studies have been performed using a with a new ORF11 construct tagged with a FLAG moiety. Results demonstrate a strong localisation to the nucleolus and also highlights colocalisation with BUD23, Extended Data Figure 13, Lines 297-300.

Figure 6: There is a need for additional data to confirm the effect on translation control. First, what is the translation efficiency and RPF density of ORF65, shown in Figure 4c,e, to be affected at the translation level? Are TE and RPF density also similarly reduced on the CDS of ORF65? The authors suggest that the effect of BUD23 knockdown is an increase in uORFs translation and consequently a reduction in translation of the main CDS. The data shown in Figure 6b is insufficient to support this notion and also kind of counterintuitive. An increase in uORF of ORF69, 75, and 61 should decrease the directly related CDS of ORF69, 75, and 61. Similarly, a decrease in translation efficiency of the main CDS of ORF28 should correlate with an increase in translation efficiency of the associated uORF. Mapping of the ribosomal reads could provide a visual information on this. The authors need to show those data to support a dysregulated KHSV transcript control that includes an increased translation of uORF and a decreased translation of main CDS caused by BUD23 depletion. Could the author provide/quantify the change in TE of uORFs and the corresponding ORFs of all KSHV genes in BUD23 depleted cells?

In that respect, what is the effect of BUD23 depletion during KHSV infection on translation of host transcripts and uORF-containing host transcripts? Can the authors look into this within their ribosome profiling data? This would be an important aspect to discuss as well.

There is no change in the translational efficiency of ORF65 in our ribosome profiling data during BUD23 depletion at a Log_2 fold change of only 0.108. As discussed above we have adjusted the conclusion for the effect of BUD23 depletion on ORF65 and K8.1 expression levels to a collapse of the KSHV late lytic cascade. This collapse does have some partial effect on late transcriptional levels, which is most likely why in the ribosome profiling experiment (Fig. 5a) we do not see a significant change in TE for ORF65 or K8.1. However, the greatest effect of BUD23 depletion on the translation of KSHV transcripts is for specific uORFs of other key late lytic transcripts. Therefore, the disruption in translation of these transcripts is what leads to the collapse of the lytic cascade and ultimate reduction in infectious virion production. It must be noted that ORF65 is used as a marker in previous experiments due to the very limited amount of KSHV late protein antibodies available.

We have now compared the TE Log_2 fold change of all KSHV uORFs, that have significantly increased translation during BUD23 depletion, with the TE Log_2 fold change of their respective CDS, including uORFs 69, 75 and 61 (Extended data fig. 16 and lines 384-386). These data demonstrate the significant increase in translation of this specific group of KSHV uORFs compared to their respective CDS during BUD23 depletion. Many annotated KSHV uORFs and their corresponding CDS are not affected by the depletion of BUD23, suggesting that depletion of BUD23 only effects KSHV uORFs with specific, currently undefined, characteristics.

From our ribosome profiling data, depletion of BUD23 did not significantly affect the translation efficiency of any human genes (Extended Data Fig. 17 and lines 386-387). Obviously, we hypothesise that BUD23 must be important for the translation of some human genes, as discussed in lines 408-420. However, we think the reason that no significant change in human genes was observed is because of the very low read depth of human genes compared to KSHV genes. The human genes only account for <8% of the mRNA reads and <30% of the ribosome footprint reads, leading to low confidence in changes in translational efficiency. A future study looking into the effect of BUD23 depletion specifically on human genes would be very interesting and shed light onto its role on human mRNA translation.

Figure 6e,f,g: Why did the author compare the shRNA1 and shRNA2 to ORF11 overexpressed cells? The luciferase expression level of shRNA 1, shRNA 2, and ORF11 expressed cells should be compared to shSCR control. There doesn't seem to be a significant effect of BUD23 depletion by shRNA 1 and shRNA 2 on luciferase translation under the 5'UTR of ORF28 and ORF34. Also the ribosome profiling data suggested that the uORF in the 5'UTR of ORF69 is induced translationally in BUD23 depleted cells, can the author validate this observation using a similar reporter assay: whether to fuse luciferase CDS right after the uORF of ORF69, or adding a small tag (HA, FLAG, etc...) fused to the 3'end of this uORF, so that uORF translation can be assessed by Western blotting.

We have now statistically compared all conditions and added all significance values to figure 5e,f,g (Previously figure 6). The reason though for comparing BUD23 depletion to ORF11 over expression in this experiment is ORF11 over expression is representing the effect seen during KSHV lytic replication and all previous experiments have used KSHV lytic replication as the baseline.

The depletion of BUD23 does not affect the translation of all KSHV uORFs and therefore we hypothesize that the context of each uORF in the 5' UTR of each gene is important for this regulation. Other assays, such as fusing uORF69 to a luciferase CDS, would disrupt the 5' UTR context and would most likely not see consistent results on translation. Furthermore, introducing a small tag to uORF69 could be interesting to investigate if the protein produced is stable and potential has functional roles in KSHV lytic replication. However, many uORFs have been described as unstable proteins which are degraded shortly after translation and therefore also do not have a functional role as the protein product.

Does the 5'UTR of ORF69, ORF28 and ORF34 alter transcription levels also? The authors should measure luciferase mRNA level (by RT-qPCR) in the experiments in panel e,f,g to rule out the transcriptional effect of the 5'UTRs.

HEK 293T cells stably expressing a non-targeting scrambled shRNA or shRNAs 1 or 2 targeting BUD23 or ORF11-FLAG exogenous expression were transfected with luciferase reported plasmids containing 5'UTRs of ORF69, 34 and 28. We performed RT-qPCR with primers specific for renilla luciferase and firefly luciferase. Results show that the 5'UTR of ORF69, 34 and 28 do not affect the transcription of the luciferase reporter gene during BUD23 depletion or with expression of ORF11 (Extended data fig. 22 and lines 420-421).

Additional question, what is the effect of ORF11 association with BUD23 on ribosome biogenesis? How does this association modify the 40S subunit, or the fully assembled ribosome, to decrease recognition and translation of KSHV uORFs/CDS?

The main role of BUD23 during 40S subunit biogenesis is to mediate the N⁷-methylation of base G1639 of the 18S rRNA. We therefore hypothesize that the increased association of BUD23 with pre-40S subunits during KSHV lytic replication drives more methylation of G1639 seen in figure 3g, lines 212-219. Furthermore, we suggest that the increased association of BUD23 with pre-40S ribosomal subunits is guided by its interaction with ORF11 (Figure 4a).

The role of N⁷-methylation of base G1639 of the 18S rRNA and its impact on translation, specifically on uORFs, is discussed in lines 499-511.

Reviewer #3 (Remarks to the Author):

This is an innovative paper in which the authors take an affinity-tag proteomic approach to determine whether KSHV infection manipulates ribosome biogenesis during its reactivation from latent to lytic phase infection.

They make a number of significant findings:

1 They find enhanced association of BUD23 with small ribosomal subunit precursor complexes during lytic KSHV infection. This appears to be critical for viral gene expression and loss of BUD23 leads to the reduced production of infectious virions.

2 They find that the previously uncharacterized KSHV lytic protein ORF11 associates with the pre-40S ribosomal subunit precursor complexes and is required for the production of new infectious virion. In the absence of ORF11 (mutated ORF11) they find reduced production of infectious virus.

This was an interesting, original and well-written manuscript. The experiments appeared to be rigorously performed and in general the data were convincing. I would be grateful if the authors could add molecular weight markers to all their protein gels to make them interpretable.

All protein gels and an RNA gel have had molecular weight markers added to them.

Is it feasible to perform a more subtle experiment with the ORF11 mutated virus? Does complementation with exogenous ORF11 rescue the ORF11 mutated virus. And if so – are they able to identify point mutation which prevent ORF11 association with the pre-40S complex?

As highlighted above in reviewer 1, point 2. Figure 4f shows that ORF11 overexpression construct in the CRISPR ORF11 deficient cell line can rescue late ORF 65 protein levels.

Identifying point mutations which prevent ORF11 association with the pre-40S complex would be very interesting and quite possibly the next step for this research project. However, due to the current lack of known direct interaction partners, or ORF11 structural information, we feel this is beyond the remit of this current manuscript.

Reviewers' Comments:

Reviewer #1:

Remarks to the Author:

In this very interesting manuscript, A. Whitehouse's group shows that during the KSHV lytic cycle specialized ribosomes are assembled to permit the translation of late viral proteins and the production of virions. This represents an innovative observation, which should be of interest to a broader community of virologists and life scientists.

The authors have comprehensively addressed the reviewers' comments and performed new experiments that confirm their earlier conclusions. I only have one major comment/suggestion:

1. In figure 4f the authors show the impact of knocking out the viral ORF11 on the production of early and late viral proteins. While the impact of knocking out ORF11 on the expression of pORF65, a late capsid protein, is clear-cut (expression of this protein is lost in two knockouts and can be rescued by over expressing ORF11), the impact on an early protein (ORF57) is not. Also, the data are contradictory: the left panel (WB) shows an INCREASE in ORF57 expression in knockdown #1, and overexpressing ORF11 in this knockdown #1 DECREASES pORF57 expression, which does not make sense. Furthermore, the quantification of this experiment in the panel on the right does not match the WB on the left in the case of the 24h time point showing the ORF57 expression. It looks as if something has been mixed up here.

Reviewer #2:

Remarks to the Author:

The authors significantly revised and improved their manuscript. The data is sound and the finding interesting. There are still some remaining questions that the authors should consider addressing when finalizing their paper for publication.

Regarding the quantification of actively translating K8.1 and ORF65 mRNAs isolated from polysome profiles of BUD23 depleted cells, the data shown in Extended Figure 9 only quantified K8.1 and ORF65 mRNA in fractions 6-9 and normalized to Scr. This can be interpreted in two ways: that there is a reduction in these mRNAs that are associated with actively translated ribosome, or that there is a global reduction in these mRNAs, hence a reduction in the relative amount of these transcripts in both the actively translating ribosome fractions as well as the poorly translating fractions. This is particularly important since BUD23 KD already was found to induce some reduction at the transcription level of K8.1 and ORF65 in Extended Figure 8, and ribosome profiling data showed no change in ORF65 TE as the authors mentioned. To rule out the later, the authors should quantify the relative mRNA level of all fractions and normalized each to the total transcript abundance from the input. With the current presented data, it is not clear that knocking down of BUD23 has an effect on translation of K8.1 and ORF65 mRNAs.

Regarding the immunofluorescence studies (supp Fig. 13) demonstrating that ORF11 localises mainly to the nucleolus and nucleoplasm in cells, indeed suggesting the modulation of ribosome biogenesis to affect downstream translation. Did the authors also observe staining of ORF11 at location of actively translating ribosome, the rough ER and in the cytoplasm? So despite showing ORF11 in pre- and early polysome fractions, are the interactions solely to reprogram ribosomes in the nucleolus but not via direct interaction with actively involved 40S ribosome in translation? Perhaps the authors could discuss this aspect further?

REVIEWER COMMENTS

We thank the reviewers for their positive comments to which we provide a point-by-point response below.

Reviewer 1.

1. In figure 4f the authors show the impact of knocking out the viral ORF11 on the production of early and late viral proteins. While the impact of knocking out ORF11 on the expression of pORF65, a late capsid protein, is clear-cut (expression of this protein is lost in two knockouts and can be rescued by over expressing ORF11), the impact on an early protein (ORF57) is not. Also, the data are contradictory: the left panel (WB) shows an INCREASE in ORF57 expression in knockdown #1, and overexpressing ORF11 in this knockdown #1 DECREASES pORF57 expression, which does not make sense. Furthermore, the quantification of this experiment in the panel on the right does not match the WB on the left in the case of the 24h time point showing the ORF57 expression. It looks as if something has been mixed up here.

Apologies, this is an error on our behalf. We have modified Fig 4f. The ORF57 blot is now correct showing that ORF11 knockdown in both cell lines have little effect on early ORF57 protein levels and densitometry is now representative.

Reviewer 2.

1. Regarding the quantification of actively translating K8.1 and ORF65 mRNAs isolated from polysome profiles of BUD23 depleted cells, the data shown in Extended Figure 9 only quantified K8.1 and ORF65 mRNA in fractions 6-9 and normalized to Scr. This can be interpreted in two ways: that there is a reduction in these mRNAs that are associated with actively translated ribosome, or that there is a global reduction in these mRNAs, hence a reduction in the relative amount of these transcripts in both the actively translating ribosome fractions as well as the poorly translating fractions. This is particularly important since BUD23 KD already was found to induce some reduction at the transcription level of K8.1 and ORF65 in Extended Figure 8, and ribosome profiling data showed no change in ORF65 TE as the authors mentioned. To rule out the later, the authors should quantify the relative mRNA level of all fractions and normalized each to the total transcript abundance from the input. With the current presented data, it is not clear that knocking down of BUD23 has an effect on translation of K8.1 and ORF65 mRNAs.

We have reanalysed the data as recommended by the reviewer, quantifying the relative mRNA levels to the input. This analysis shows a similar finding to the previous analysis we used, confirming a reduction in the translation of late viral proteins, whereas no decrease is observed with early protein production. New Extended Figure 9.

Regarding the immunofluorescence studies (supp Fig. 13) demonstrating that ORF11 localises mainly to the nucleolus and nucleoplasm in cells, indeed suggesting the modulation of ribosome biogenesis to affect downstream translation. Did the authors also observe staining of ORF11 at location of actively translating ribosome, the rough ER and in the cytoplasm? So despite showing ORF11 in pre- and early polysome fractions, are the interactions solely to reprogram ribosomes in the nucleolus but not via direct interaction with actively involved 40S ribosome in translation? Perhaps the authors could discuss this aspect further?

This is an interesting suggestion, we did not see any localisation with ORF11 at the ER/cytoplasm, suggesting the primary role of ORF11 affects earlier stages of ribosome biogenesis in the nucleolus/nucleoplasm. We have referenced this in the discussion (Discussion lines 356-361).